**Subject Category:**
Biology (whole organism)

cognition/behaviour/evolution

prosocial behaviour, inequity, reciprocity, prosocial choice test, African grey parrot
(*Psittacus erithacus*)

**Authors for correspondence:**
Anastasia Krasheninnikova
e-mail: akrashe@orn.mpg.de
Auguste M. P. von Bayern
e-mail: avbayern@orn.mpg.de

# Assessing African grey parrots' prosocial tendencies in a token choice paradigm

Anastasia Krasheninnikova[1,2], Désirée Brucks[1,2],
Sigrid Blanc[2,3] and Auguste M. P. von Bayern[1,2,4]

[1]Max Planck Institute for Ornithology, Eberhard-Gwinner-Strasse, 82319 Seewiesen, Germany
[2]Max Planck Comparative Cognition Research Station, Loro Parque Fundacíon, 38400 Puerto de la Cruz, Tenerife, Spain
[3]Laboratoire d' Ethologie Expérimentale et Comparée, EA 4443, Université Paris 13, Villetaneuse, France
[4]Department of Biology, Ludwig-Maximilians-University of Munich, 82152 Planegg-Martinsried, Germany

AK, 0000-0001-8566-8277; DB, 0000-0003-3146-5110

Prosociality is defined as a voluntary, typically low-cost behaviour that benefits another individual. Social tolerance has been proposed as a potential driver for its evolution, both on the proximate and on the ultimate level. Parrots are an interesting species to study such other-regarding behaviours, given that they are highly social and stand out in terms of relative brain size and cognitive capacity. We tested eight African grey parrots in a dyadic prosocial choice test. They faced a choice between two different tokens, a prosocial (actor and partner rewarded) and a selfish (only actor rewarded) one. We found that the birds did not behave prosocially when one subject remained in the actor role; however, when roles were alternated, the birds' prosocial choices increased. The birds also seemed to reciprocate their partner's choices, given that a contingency between choices was observed. If the food provisioned to the partner was of higher quality than that the actor obtained, actors increased their willingness to provide food to their partner. Nonetheless, the control conditions suggest that the parrots did not fully understand the task's contingencies. In sum, African grey parrots show the potential for prosociality and reciprocity; however, considering their lack of understanding of the contingencies of the particular tasks used in this study, the underlying motivation for the observed behaviour remains to be addressed by future studies, in order to elucidate the phylogenetic distribution of prosociality further.

# 1. Introduction

Prosocial behaviour is usually defined as a voluntary action that benefits another individual without incurring any immediate gains to the actor (nor involving any substantial cost like an altruistic behaviour) [1]. Prosociality has recently gained much attention as a driving mechanism for the evolution of complex cooperation in human societies [2]. Experimental evidence indicates that prosociality in humans is motivated, at least in part, by empathy [3] and concern for the welfare of others ([1]; hereafter referred to as other-regarding preferences). To understand the evolutionary origins of human prosociality, researchers have thus focused on non-human primate species presuming they are our closest living relatives. However, the results show no clear link between phylogenetic closeness to humans and their higher prosocial tendencies (see [4] for a review). In fact, there is evidence that capuchin monkeys, but not chimpanzees [5,6], avoided selfish choices when they could procure food for a neighbouring partner at the same time without extra effort [7].

Several hypotheses concerning the underlying motivation of prosociality in non-human primates have been proposed. At the ultimate level, kin selection delivers indirect fitness benefits to related individuals, whereas reciprocity (where prosocial behaviours are conditional upon having received similar behaviours from others previously) plays a role in the maintenance of altruistic-like behaviours between unrelated individuals, provided that such acts benefit the donor's inclusive fitness [8–10]. At the proximate level, different mechanisms, such as empathy, have been suggested to underlie prosocial behaviours [3,11,12]. Others have proposed that prosocial behaviour could result from high levels of social tolerance [13], a strategy to avoid harassment from conspecifics [14,15] or convergent evolution among cooperative breeders [16].

Indeed, cooperative behaviours in non-human primates seem to be directed to kin and to familiar, hence reciprocating, partners [17,18], but have also been reported between non-kin partners and in situations without a possibility for reciprocity [19,20]. For example, marmosets (*Callithrix jacchus*) were more prosocial towards kin than non-kin conspecifics [16], and cottontop tamarins (*Saguinus oedipus*) provided more food to their familiar cage-mates than strangers [21]. Similarly, capuchin monkeys (*Cebus apella*) demonstrated sensitivity to others' welfare [7] and acted prosocially towards group members but not to strangers [19]. They also increased their prosocial behaviour in a situation of direct reciprocity, thus showing high flexibility in their prosociality [22]. However, the results of primate studies are inconsistent. For example, chimpanzees do [23] or do not [5,24] show prosocial behaviour depending on the tests used. Given these inconsistencies and the various hypotheses that have been proposed to explain the driving forces of prosocial behaviour, it is helpful to broaden the phylogenetic spectrum of species studied. A comparative approach may enhance our understanding not only of the socio-ecological living conditions but also the underlying mechanisms that may have favoured the selection for prosocial tendencies [4], particularly if the studies on different species are based on similar paradigms [25]. In the past decade, the study of prosociality has been extended to other non-primate and even non-mammalian taxa but using many different methodologies [25].

The prosocial choice task (PCT) is probably the most frequently used method to measure prosociality [6,20]. The basic principle of the PCT requires subjects to choose between two options of different value, namely either an option (a token or a bar) that rewards both actor and partner (prosocial choice: 1/1) or an option that results in food only for the actor (selfish choice: 1/0) [26,27]. Such a paradigm allows the subjects to reward their partner without a supplementary cost for themselves. As PCT has been used in various species such as rats [26], dogs [28] and birds [29], it, therefore, is a valuable task to be adopted for providing further comparative data.

Members of the corvid family have become an important and good model for a comparative approach for studying prosociality, as their socio-cognitive abilities rival those of primates [30–32], suggesting convergent evolution of those traits [31,32]. Similar to the findings on primates, those on corvid prosociality are mixed without a clear detectable pattern yet. In studies using other paradigms, jackdaws (*Corvus monedula*), for example, show active food sharing [33–35] and also deliver food to their partner when given the choice between prosocial and a selfish choice [36]. Pinyon jays provide food to a partner in a PCT, however, only if they were also rewarded; when using a more costly set-up that rewarded only their partner but not themselves, they acted randomly [37]. By contrast, ravens (*Corvus corax*) did not behave prosocially towards partners, regardless of their relationship [38] or the experimental paradigm employed [29].

Parrots are another group of large-brained birds whose cognitive and communicative abilities also parallel those of primates [39], making them interesting avian candidates to investigate prosociality. African grey parrots (*Psittacus erithacus*), as highly social species, lend themselves particularly well to

the investigation of their prosocial tendency, given that high levels of tolerance have been proposed to be an important condition under which prosociality can evolve [40,41]. African grey parrots have been observed in flocks of up to 1200 individuals [42,43], which split up during the day for foraging and reunite again during the night, hence exhibiting fission–fusion dynamics [44]. Furthermore, they probably form longer-term monogamous pairs and exhibit biparental care, i.e. males assist in feeding offspring [43]. Moreover, it has been observed that they collectively mob invaders [45]. Although few studies have focused on African grey parrot behaviour in the wild, a large number of studies have investigated their cognitive abilities in the laboratory (see [46] for a review). This research suggests that they possess enhanced cognitive abilities, even described as similar to those of young children [47,48]. Naturally, African grey parrots exhibit a range of affiliative behaviours (e.g. regurgitation of food for sharing, mutual affiliative interaction such as grooming; [42]), and experimental studies revealed that they are able to coordinate their actions in an artificial string-pulling task [49] and that they respond to an experimenter's communicative cues and attentional states [50–52]. However, while there is a continuously growing number of studies on prosocial behaviour in corvids (see above), such studies are lacking in parrots with the exception of two studies by Péron *et al.* [53,54]. In the first study, two African grey parrots were tested in a dyadic setting, in which they alternately chose between four options that could reward both individuals, either individual or none of the birds [53]. The results suggested that the birds were able to discriminate between the four options and that they began to adapt their choices in response to their partner's choices. While the parrots did not show clear reciprocity, they changed their behaviour based on who started as the actor in each session. If the dominant bird started the session first, he chose the prosocial token but became more selfish after he experienced selfish behaviour from his partner. When the dominant bird was the follower, he retained his prosociality. The subordinate bird, however, became more selfish over time in both roles. Therefore, both birds developed a selfish tendency over time. In the second experiment, the parrots exchanged with a selfish or generous human partner and tended to match their choices to the respective human partner [53]; however, this study had a few limitations, in particular in controlling for side biases. Even though the authors had carried out a procedure to minimize the formation of a side bias, by switching the order of the cup position, this procedure was carried out only twice per subject (i.e. Experiment 1: two position changes interspersed among the first nine sessions; Experiment 2: once per session (see electronic supplementary material, figure S1 in [53]); consequently, it remained unclear whether the parrots' choices were based on preferences for certain cup positions or indeed on prosocial tendencies. In their later follow-up experiment, in which one of the previously tested birds exchanged tokens with a human experimenter who exactly mirrored the parrot's behaviour, the parrot did not show any side preferences though [54]. Furthermore, both studies were compromised by their small sample size (i.e. $N = 2$ in [53], $N = 1$ in [54]) and potential colour preferences that were not controlled for (see also [25]).

The objectives of this study were threefold, first, to contribute to the field of comparative cognition research by providing more data on prosocial tendencies in African grey parrots (*P. erithacus*) implementing several methodological improvements suggested by earlier studies. Our second objective was to achieve a more direct comparability of the grey parrots' performance to that of primate species by employing a token exchange PCT that was similar to that employed by earlier studies on primates [20,22] and, third, to examine whether the birds would develop reciprocity when provided with a possibility for reciprocation with their conspecifics. We incorporated methodological changes outlined by Tan *et al.* [41] as well as Marshall-Pescini *et al.* [25], which meant to ensure that subjects would pay attention to the outcome of their actions. Furthermore, we incorporated several control conditions which may assess the animals' understanding of the task and potential effects of social facilitation, while at the same time avoiding the pitfalls of overtraining the subjects' behavioural response (reviewed in [25]). Additionally, we varied the reward distribution, using 'equal' or 'unequal' rewards, to test whether unequal reward distribution (i.e. making a prosocial choice that would provide the recipient with higher quality food than the donor) affected prosocial choices (see Material and methods for details). This latter condition was implemented since previous studies have shown that inequity can lead to the reduction or termination of cooperation between individuals [55]. Considering African grey parrots' high level of social tolerance, we predicted that (i) they would show a spontaneous tendency to benefit a partner (unilateral condition) and (ii) this tendency may be further enhanced by the possibility for reciprocal actions when alternating the subjects' roles trial by trial (alternating condition). We further expected that (iii) birds would reduce their willingness to choose the prosocial token when facing inequity (in favour of the recipient).

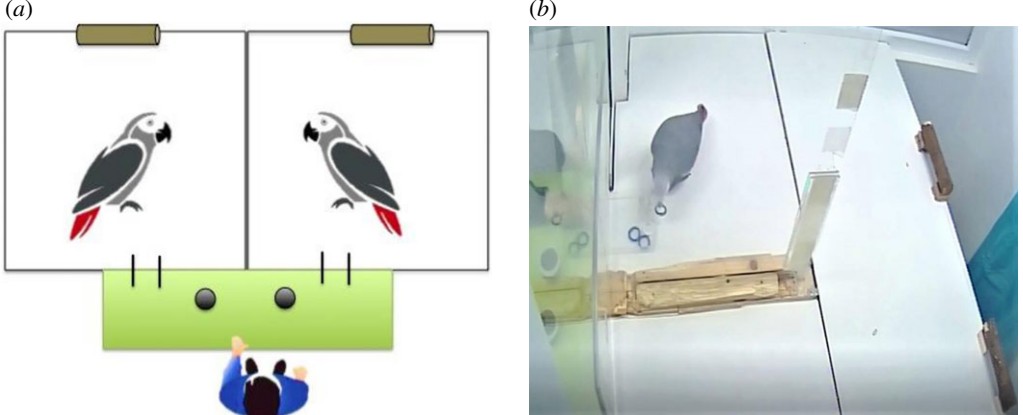

**Figure 1.** (*a*) Schematic overview of the experimental set-up with the two birds in adjacent compartments (not to scale) and the experimenter opposite. The dividing wall between the birds as well as the divider between the birds and the experimenter were made of transparent Plexiglas. The latter contained two exchange holes so that both birds could exchange tokens against food with the experimenter who faced them sitting at a table onto which two cups that contained the respective rewards were placed. Each compartment contained a white table serving as floor at the height of the experimenter's experimental table and had a perch located at its back. The remaining test room behind the table was shut off by a curtain to prevent birds from flying to the other side. (*b*) Picture of the test chamber set-up in the accessible control condition. Here, an additional table was placed behind the perches and dividing wall, thus connecting both chambers.

## 2. Material and methods

### 2.1. Subjects and housing

The study was conducted from March to June 2018 with eight subadult African grey parrots (see electronic supplementary material, table S1 for individual characteristics). The parrots were hand-raised, but group-reared, in the Loro Parque Fundación, Tenerife, Spain. They were housed together in an outdoor aviary (21.41 m$^2$) equipped with perch structures and wood toys in view of the zoo visitors, in the Loro Parque zoo, Puerto de la Cruz, Tenerife. All birds always had free access to water and mineral blocks and were fed fresh fruits and vegetables twice a day. Parrots were not food deprived, but the daily amount of nuts and seeds (Versele-Laga Loro Parque Ara Mix) provided to the birds during feeding times was adjusted according to their intake during testing for weight control purposes. All birds were familiar with the testing routine (i.e. transport to test room and test room itself) from previous studies [56,57].

### 2.2. Set-up

Training and testing took place in an indoor room sized 2.5 × 1.5 × 1.5 m equipped with lamps covering the birds' full range of visible light (Arcadia 39 W Freshwater Pro and Arcadia 39 W D3 Reptile lamp). A sound-buffered one-way glass system permitted zoo visitors to view the experiment conducting inside the room but not the birds to see out.

The test chamber had one table (87 cm height × 49 cm width × 150 cm length) and two perches placed at the back of the table (figure 1*a*). For all sessions (except for the familiarization and the token preference tests), the table was divided into two equally sized compartments (150 × 49 × 75 cm) by a transparent Plexiglas sheet; this allowed the birds to see and hear each other but preventing them from direct contact. A curtain was used to divide the rest of the room. For the accessible condition (see description below), a second table was added behind the perches of the first one so that the birds could walk into the other compartment (figure 1*b*). The front of the test chamber consisted of a Plexiglas wall with two 5 × 5 cm holes at 10 cm height in the middle of the two compartments through which the parrots could exchange tokens with the human experimenter who sat in front of them in another room (see electronic supplementary material, Video S1). Subjects were tested either individually or in pairs, depending on the test condition (see table 1 for an overview of conditions).

Six metal rings (each 2.5 cm in diameter) were used as non-edible tokens. Three metal rings were covered with electrical tape of a colour (e.g. blue) and another three were covered with another colour

**Table 1.** Summary of test and control conditions per bird with the role of partner and number of test sessions (including number of trials) and respective predictions. active, actively selecting tokens and receiving rewards; inactive, not selecting tokens but only receiving rewards; absent, not present in test room. S, selfish token; P, prosocial token.

| test order | | reward distribution | partner | description | sessions (no. of trials) | predictions |
|---|---|---|---|---|---|---|
| | *test conditions* | | | | | |
| 1 | unilateral (UNI) | equal<br>unequal | inactive | actor chooses tokens | 2 (30 trials)<br>2 (30 trials) | P > S |
| 2 | alternating (ALT) | equal<br>unequal | active | actor and partner take turns in selecting tokens | 4 (2 per bird starting as actor; 15 trials each) | P > S (more P than in UNI) |
| | *control conditions* | | | | | |
| 3 | yoked (YC) | equal<br>unequal | inactive | actor and experimenter alternately select tokens; experimenter copies partner's choices from previous ALT session | 2 (15 trials)<br>2 (15 trials) | S > P (less P than in ALT) |
| 4/5[a] | accessible (ACC) | equal<br>unequal | absent | actor can access partner's compartment | 1 (30 trials)<br>1 (30 trials) | P > S |
| 4/5[a] | inaccessible (INACC) | equal | absent | actor cannot access partner's compartment | 1 (30 trials) | S > P/S = P |
| 6 | social facilitation (SFC) | equal | inactive | actor selects tokens but partner cannot access them (front panel is blocked) | 1 (30 trials) | S > P/S = P |

[a]The non-social conditions were presented in random order.

(e.g. white) in order to create two easily distinguishable tokens. The tokens used were differently coloured for each value (prosocial/selfish) and reward distribution (equal/unequal). Each dyad had a unique set of token colours (see electronic supplementary material, table S2). Furthermore, colour preferences were assessed prior to training (see §2.3.1). The values of tokens were randomly assigned: the selfish token provided a reward only to the actor (1/0), while the prosocial token yielded a reward to both actor and partner (1/1). This procedure allowed the actor to obtain a seed, independent of the token choice in all conditions. When the actor chose a prosocial token in the equal reward distribution, both birds received sunflower seeds without shells. When the actor chose a prosocial token in the unequal reward distribution, the actor received sunflower seeds whereas the partner received a piece of walnut (preference for walnut over sunflower seeds was assessed as a highly preferred food reward for these parrots prior to the experiment).

## 2.3. General procedure

The birds were tested once or twice a day, in the morning and/or afternoon. To ensure sufficient motivation, tests took place only within a minimum of 3 h after the last feeding or before feeding in the case of morning sessions. All subjects were already familiar with the token exchange procedure from previous studies [57,58]. Each subject was paired with a conspecific partner with whom it maintained a strong affiliative bond based on previous observational data prior to the study. Of the four pairs tested, two were same-sexed siblings and two were opposite-sexed (see electronic supplementary material, table S1). In each condition, the subject (=actor) exchanged tokens with the experimenter, while the other bird served as the passive partner, except for the alternating condition, in which both birds took turns in exchanging tokens with the experimenter (see description below). Prior to testing, each bird was randomly assigned to one fixed compartment side throughout the experiment to make sure that the subjects 'knew' their role (i.e. who is the individual that makes the choice) across the different conditions.

The birds had also previously experienced a similar set-up, in which they exchanged tokens with an experimenter, while they had a conspecific in a neighbouring chamber. However, the procedure of that latter study, which tested the birds' response to inequity [58], differed from the study presented here, as the subject could not influence what themselves and the neighbour would receive from the experimenter, and could not choose between different tokens (see Discussion).

To avoid the formation of location biases during token selection, the tokens (six in total—three of each colour) were jumbled in a small cup and then randomly placed onto a shallow transparent plastic tray (making sure that the tokens did not overlap but were still randomly distributed across the tray, i.e. without predefined position for six tokens) before presenting them to the birds. In each trial, the experimenter presented the tokens on the tray and pushed it through the slot underneath the chamber window of the actor's compartment (see electronic supplementary material, Video S1) where it could choose a token. Meanwhile, the partner could witness the actor's behaviour at all times. Once the actor had made its choice (i.e. had lifted one of the six tokens), the tray with the remaining tokens was immediately removed by the experimenter.

The experimenter requested the token back with a specific gesture holding the hand outstretched with the palm turned upwards in front of the exchange hole and calling the bird's name. The bird had 15 s for exchanging the token. If it did not exchange the token within this time or failed to place the token in the experimenter's hand (e.g. throwing the token to the floor), the trial was terminated and repeated followed by a longer inter-trial interval (30 s). Once the bird had exchanged the token (i.e. placed it in the experimenter's hand), the experimenter placed the token on the table in a position visible to both birds. The food rewards that were contained in two small opaque plastic cups (5 cm diameter, 4 cm height) each covered with a lid were present in front of each compartment during each trial (figure 1a). The rewards inside the cups were not visible to the birds. Once a choice was made, the experimenter removed the lid(s) from the small food container(s), tilted the cup(s) forward so that both birds could see the reward(s) inside, and then distributed the reward(s) (see electronic supplementary material, Video S1). In the case of a prosocial choice, the experimenter removed both lids of the food containers at the exact same time and rewarded the actor 1–2 s before the partner so that the latency between returning a token and receiving a reward remained the same for the actor irrespective of its choice (see electronic supplementary material, Video S1 for the procedure). In the case of a selfish choice, the experimenter opened and tilted only the actor's container before delivering the rewards. Following a 10 s inter-trial interval, the next trial began. Session length and number of trials per session varied according to the type of distribution (see table 1 for an overview) and lasted approximately 30 min. Each test session

could be shortened or extended depending on the birds' motivation; however, 30 trials (i.e. a maximum of 30 rewards) were never exceeded (excluding tiny rewards, i.e. max. half of a seed that was used to move some individuals back into their starting position; it happened in all birds in around 10% of the trials). If the actor failed to return the token in more than five consecutive trials in a session, the session was terminated and repeated on the following day (12 out of 72 sessions needed to be repeated across all dyads in total for different reasons; see electronic supplementary material for further information).

### 2.3.1. Token preference tests

Before the beginning of the study, the birds had the opportunity to interact with the tokens. To ensure that the birds had no colour preferences prior to testing, we conducted token preference tests before the token association training sessions. Token preference tests were carried out separately for the tokens used in the equal and unequal reward distribution (always before the association training). Each bird could select a token from a shallow tray containing two differently coloured tokens. The tokens were presented in a pseudo-randomized and counterbalanced way in four predefined token positions on the tray (i.e. left–right or front–back). After the bird chose one token, it was rewarded with a sunflower seed (independent of the selected token) and the trial ended. The next trial started after a 10 s interval. Following the methods by de Waal et al. [19], each subject was offered 12 choices, and if any token was chosen 10 times or more by one of the two birds of one pair, the token would be replaced by another colour in the next session (this happened between one and three times per bird). The purpose of this test was to ensure that the birds' choices were not biased by preferences of a specific colour and to establish a set of token colours for each pair of birds for the rest of the experiment (electronic supplementary material, table S2).

### 2.3.2. Token familiarization

In order to familiarize the birds with the different token values, they had to complete a token familiarization phase. The actor and the partner were present during the sessions but only the actor could participate and pick up the tokens. Two sessions per bird were conducted (four sessions per dyad) on either the same day or two consecutive days, if necessary. Thirty 'forced' choice trials were conducted per session (i.e. 15 with only the selfish (1/0) token and 15 with the prosocial (1/1) token). With the help of an online random number generator (random.org), we determined the order of trials (selfish/prosocial, see description above for rewarding schemes). The subjects received the two token familiarization sessions prior to being in the role of the actor in the two unilateral sessions (i.e. one before the equal reward distribution and one before the unequal reward distribution).

### 2.3.3. Test conditions

Six different conditions (unilateral (UNI), alternating (ALT), yoked control (YC), accessible control (ACC), inaccessible control (INACC), social facilitation control (SFC)) were tested in the present study (see table 1 for an overview). We closely followed the procedure of Suchak & de Waal [22] and de Waal et al. [19]. The parrots received two sessions of 30 trials per condition for the first three conditions.

The first actor had to complete the token familiarization phase and then the unilateral condition before the roles were reversed for the following sessions during which the partner had to complete those same sessions. Following this, the dyads passed on to the alternating condition. The unilateral, alternating and yoked control conditions were conducted first with an equal and subsequently with an unequal reward distribution (see electronic supplementary material, table S5 for an overview and the testing order of conditions given as an example for the dyad Nikki–Jack). Before proceeding with testing the birds in those conditions with unequal reward distribution, the birds received one additional accessible session where they could obtain a second (unequal) reward to ensure that the birds understood the token values (see electronic supplementary material, table S5).

#### 2.3.3.1. Unilateral condition

The unilateral conditions tested for spontaneous prosociality without the possibility for immediate, direct reciprocity. One bird (actor) could choose tokens throughout 30 trials, while the partner could only witness the choices (and receive rewards in the case of a prosocial choice) but not interfere (see electronic supplementary material, Video S1). Two sessions with 30 trials each (in total 60 trials) were conducted before switching the roles. Before switching to the actor role, the partner received two token familiarization sessions (see description above). The same procedure was followed when the conditions were repeated with unequal reward distribution.

### 2.3.3.2. Alternating condition

In order to examine if the parrots showed reciprocity in their choices, an alternating condition was incorporated. Contrary to the unilateral condition, the birds could choose tokens alternately (see electronic supplementary material, Video S1). Two sessions with 30 trials each were conducted (in total 30 trials per bird). The actor made the first choice throughout those two sessions. Following the alternating condition, the yoked control was carried out for this initiating actor (see below for description) before the roles were reversed and the previous partner was now allowed to initiate the alternating condition (=make the first choice throughout two consecutive sessions). This was followed by the respective yoked control for this bird (see electronic supplementary material, table S5 for an overview).

### 2.3.3.3. Yoked control condition

In order to understand whether the birds were responding to their partner's behaviour (i.e. behaving prosocial or not) in the alternating sessions rather than just to the rewards they received, the yoked control condition was implemented. While both birds were present, only the actor was allowed to make a choice every other trial (just as in the alternating sessions). The partner was not allowed to make a choice but instead, the experimenter copied the partner's choice from each trial from the previous alternating session, i.e. the experimenter—visible from both compartments—chose the token from the tray (see below). Consequently, the partner's choices were exactly replicated by the experimenter (i.e. 'yoked'), while the partner was present only passively. The partner bird was not able to choose since the tray with the tokens was not moved into its compartment and remained in front of the two compartments. This means that the actor received the same number of rewards as in the previous alternating session; however, this payoff was not a result of the partner's behaviour. Hence, if the subject understood that the second individual could not choose, it should not develop reciprocity/respond in a reciprocal manner in this control and behave differently compared to the alternating condition. When the experimenter had to reproduce the previous choice of the second individual, she placed the token box in the middle of the table and chose one token in front of the birds, placed it on a clearly visible spot on the table in front of the chamber window between both birds (see electronic supplementary material, Video S1). If the prosocial token was displayed, the experimenter gave the first reward to the partner (i.e. the 'passive' actor who did not participate but had made this particular choice in the corresponding trial in the previous alternating session as the actor) and then rewarded the actor in the partner role after a 1–2 s delay.

### 2.3.3.4. Accessible control

In this control, the actor chose without a partner being present while having access to the neighbouring compartment. It could thus obtain a second reward when making a 'prosocial' choice. This test examined whether the subjects had understood that the prosocial token delivered a reward to each test compartment. Contrary to Suchak & de Waal [22], we called this condition 'accessible control' instead of 'open-panel control', since we did not open the panel in-between compartments but rather added a second table at the back of the test room (figure 1*b*). This allowed the birds access to both compartments by walking on the table from one to another compartment. Prior to the partner-absent 'accessible' control, the birds received a familiarization session so they would know about the second table at the back of the compartment, that gave the subject access to both compartments. The familiarization session also ensured that the birds were fully habituated to the second table and knew that they could access both compartments freely. We gave the birds 12 trials in which we visibly placed the reward in the respective (empty) neighbour compartment. After the bird entered the other compartment and consumed the reward, the next trial began with the next reward being placed in the compartment on the other side. The position of the reward thus alternated between the two compartments upon every trial, with each compartment baited in an equal number of trials (a reward was presented six times in the left and six times in the right compartment). The subjects had to change between the compartments and consume the offered rewards throughout all trials, before the actual partner-absent 'accessible' control condition was proceeded.

Procedures followed those of the unilateral sessions except that there was no partner present. As before, in the case of a prosocial choice, the reward was delivered first to the actor and after the usual delay (as in the other social conditions) to the empty compartment (see electronic supplementary material, Video S1). The second reward in the empty compartment was removed after 15 s if the actor had not entered and consumed it until then (this happened only for one bird (Jelo) in 84% of prosocial choices). The next trial began after 10 s. The order of the non-social control conditions was

counterbalanced between individuals with half of the birds starting with the accessible control and the other half with the inaccessible control.

### 2.3.3.5. Inaccessible control

The inaccessible control assessed the birds' choice behaviour in the absence of their partner in order to examine whether the birds flexibly changed their choices according to the condition rather than consistently choosing the prosocial token (having formed a habit or preference for it). If the birds differentiated between the conditions, they should return to random choice between the tokens in this condition; whereas, they should choose the prosocial token in the accessible condition. In this control, the bird could not access the partner compartment; the configuration of the compartment was the same as that of the other testing conditions (see electronic supplementary material, Video S1). In the case of a prosocial choice, the second reward was delivered to the empty compartment as in the previous accessible control; however, it was removed again after 2–3 s.

### 2.3.3.6. Social facilitation control

The social facilitation control allowed us to investigate if the mere presence of a conspecific affected the actor's motivational state to choose prosocially. As in the test conditions, the actor was allowed to choose tokens in the presence of the partner in the neighbouring compartment. However, in this control, the partner was unable to access any food, even if a prosocial choice was made; the partner's exchange hole was closed and conspicuously covered with a black plastic panel (see electronic supplementary material, Video S1). Therefore, if the actor selected a prosocial token, it was rewarded normally while the reward of the partner was not delivered. Instead, it was placed in front of the blocked panel for 2–3 s and then removed.

## 2.4. Analyses

All videos were recorded with four cameras from different angles. The parrots' choices were directly noted during the experiment. Analyses were conducted in R [59] using the package 'lme4' [60]. We used generalized linear mixed models (GLMM) with a binomial distribution to analyse the frequency of prosocial choices across test conditions. Accordingly, we set the number of prosocial and selfish token choices as the response variable and test conditions (factor: UNI, ALT, YC, ACC, INACC, SFC) as well as reward distribution (factor: equal, unequal), as fixed effects. Individual identity and dyads were entered as random effects. In order to assess potential changes in token choices during the course of a test session, we analysed choices across 10-trial blocks in a separate GLMM including an interaction between blocks (continuous: 1–6) and conditions as fixed effects. Moreover, in order to analyse the contingency between choices in the alternating conditions (equal and unequal), we calculated $2 \times 2$ $\chi^2$ contingency tables [22].

# 3. Results

## 3.1. Token preferences

None of the birds showed any token preferences prior to the test (two-sided binomial: all $p > 0.39$); however, when retesting the birds' preferences following the token exchange task, we found that some birds had developed preferences. Two birds exhibited a significant preference for either token used in the equal reward distributions (one for prosocial token/one for selfish token; two-sided binomial: $p < 0.02$), while four birds showed preferences for either token used in the unequal reward distribution (two for prosocial token/two for selfish token; binomial: $p < 0.02$; see electronic supplementary material, table S2 for individual results). Considering that the distribution of token preferences was balanced at the group level, we did not further control for it in the subsequent analyses.

## 3.2. Conditions

When comparing conditions based on whether equal and unequal rewards were distributed, we found that the birds chose the prosocial option more often during the UNI condition when unequal rewards were delivered compared to equal rewards (GLMM: $\beta = 1.00$, s.e. $= 0.23$, $z = 4.41$, $p < 0.001$; figure 2).

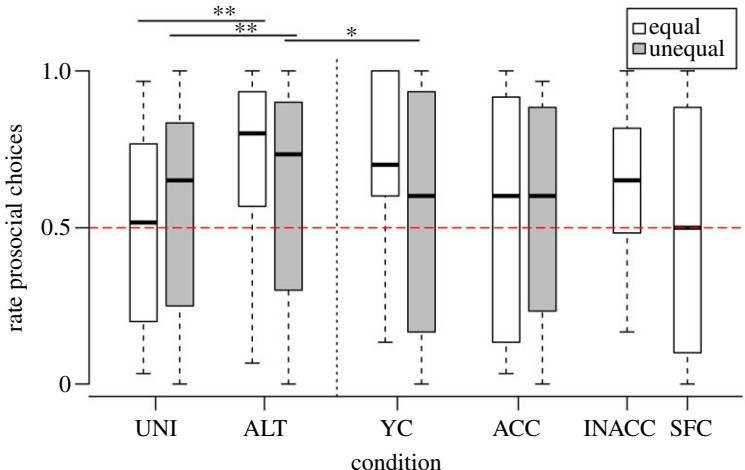

**Figure 2.** Proportion of prosocial choices (group performance) across test conditions separately for equal (white) and unequal rewards (grey). The dashed red line represents chance level (0.5). UNI, unilateral condition; ALT, alternating condition; YC, yoked control; ACC, accessible control: INACC, inaccessible control; SFC, social facilitation control. *$p < 0.05$; **$p < 0.01$.

**Table 2.** Contingency table of all individuals' choices in the alternating conditions using equal rewards as function of the partner's choice in the previous trial. Percentages are depicted in parentheses (see electronic supplementary material, table S4 for contingency tables for each dyad).

| actor's choice at trial $n$ | partner's choice at trial $n-1$ | | |
|---|---|---|---|
| | selfish | prosocial | total |
| selfish | 37 (48.1) | 40 (51.9) | 77 (100.0) |
| prosocial | 32 (19.6) | 131 (80.4) | 163 (100.0) |

No difference was found in the YC using equal and unequal rewards (GLMM: $\beta = -0.49$, s.e. = 0.29, $z = -1.73$, $p = 0.084$). Moreover, the birds tended to choose the prosocial token less frequently during the ALT condition, when unequal rewards were delivered compared to equal rewards (GLMM: $\beta = -0.20$, s.e. = 0.11, $z = -1.83$, $p = 0.068$). Since the full model containing both equal and unequal reward distributions showed a significant interaction between condition and reward type (equal/unequal; $\chi^2 = 36.19$, d.f. = 3, $p < 0.001$), we split the data up and analysed the equally and the unequally rewarded conditions separately.

### 3.2.1. Equal reward distributions

The birds chose the prosocial token significantly more often than chance level (0.5) in the ALT and YC conditions (one-sample $t$-test: ALT: $t = 3.81$, d.f. = 31, $p < 0.001$; YC: $t = 3.01$, d.f. = 15, $p = 0.009$), while their choices did not differ from chance level in the UNI, SFC, INACC and ACC conditions (one-sample $t$-test: UNI: $t = -0.15$, d.f. = 15, $p = 0.880$; SFC: $t = -0.11$, d.f. = 15, $p = 0.917$; INACC: $t = 2.33$, d.f. = 15, $p = 0.034$, n.s. after Bonferroni's correction; ACC: $t = 0.45$, d.f. = 15, $p = 0.657$).

We found a main effect of condition on the birds' token choices ($\chi^2 = 113.81$, d.f. = 5, $p < 0.001$; figure 2). Accordingly, the birds chose the prosocial option more frequently in the ALT condition (mean ± s.d.: 69.8 ± 29.4%) compared with the UNI condition (48.8 ± 32.5%; GLMM: $\beta = 1.21$, s.e. = 0.16, $z = 7.61$, $p < 0.001$). To understand whether the increased prosocial behaviour resulted from reciprocity or reinforcement learning, we compared the birds' prosocial choices in the ALT condition (69.8 ± 29.4%) with the choices in the YC condition (71.3 ± 28.3%). We found no difference between those two conditions (GLMM: $\beta = 0.09$, s.e. = 0.20, $z = 0.46$, $p = 0.644$). In order to find out whether the birds based their token choices on their partner's choice in the previous trial, we tested the contingency between the selected tokens in the ALT condition, and found a strong contingency between choices ($\chi^2 = 131.93$, d.f. = 1, $p < 0.001$; table 2).

**Table 3.** Contingency table of all individuals' choices in the alternating conditions using unequal rewards as function of the partner's choice in the previous trial. Percentages are depicted in parentheses (see electronic supplementary material, table S4 for contingency tables for each dyad).

| | partner's choice at trial $n-1$ | | |
|---|---|---|---|
| actor's choice at trial $n$ | selfish | prosocial | total |
| selfish | 59 (59.0) | 41 (41.0) | 100 (100.0) |
| prosocial | 22 (15.7) | 118 (84.3) | 140 (100.0) |

The birds' prosocial choices did not differ between the UNI condition and the SFC condition (49.5 ± 39.2%; GLMM: $\beta = 0.01$, s.e. = 0.15, $z = 0.08$, $p = 0.940$); consequently, we cannot rule out that the presence of the partner facilitated prosocial choices irrespective of whether it helped the partner or not. The birds chose the prosocial token more frequently in the INACC condition compared with the ACC condition (54.2 ± 36.8%; GLMM: $\beta = 1.71$, s.e. = 0.49, $z = 3.47$, $p < 0.001$). We can rule out an order effect (i.e. experiencing INACC condition before ACC condition and vice versa) on the occurrence of prosocial choices in the controls ($\chi^2 = 2.67$, d.f. = 1, $p = 0.102$). In the absence of a partner (i.e. INACC condition), birds chose the prosocial option significantly more often than during the corresponding UNI condition (GLMM: $\beta = 0.88$, s.e. = 0.16, $z = 5.67$, $p < 0.001$), and also more often than in the SFC condition in which their partner could not be able to access the rewards (GLMM: $\beta = -0.87$, s.e. = 0.16, $z = -5.60$, $p < 0.001$). This effect might be explained by the fact that the birds changed their token choices throughout test conditions (i.e. condition × 10-trial blocks interaction: $\chi^2 = 13.81$, d.f. = 5, $p = 0.017$). In particular, the birds' behaviour in the INACC condition in which the birds decreased their choice for the prosocial option across the six 10-trial blocks (GLMM: $\beta = -0.22$, s.e. = 0.09, $z = -2.44$, $p = 0.015$) drove this effect. Birds did not change the choice of the prosocial option during the course of the other conditions (GLMMs: UNI: $\beta = 0.05$, s.e. = 0.09, $z = -0.50$ $p = 0.615$; ALT: $\beta = 0.05$, s.e. = 0.09, $z = 0.50$, $p = 0.615$; SFC: GLMM: $\beta = -0.08$, s.e. = 0.09, $z = -0.50$, $p = 0.615$; YC: $\beta = -0.05$, s.e. = 0.21, $z = -0.22$, $p = 0.825$; ACC: $\beta = 0.01$, s.e. = 0.09, $z = 1.11$, $p = 0.266$).

### 3.2.2. Unequal reward distribution

A preference test prior to the unequal test conditions revealed that walnut was consistently preferred over sunflower seeds (two-sided binomial $p < 0.038$), and with the exception of two birds that needed two sessions, all birds exhibited this preference during the first presentation.

In the unequal conditions, the birds' prosocial choices did not differ from chance level (0.5) in any of the conditions (one-sample $t$-test: UNI: $t = 0.73$, d.f. = 15, $p = 0.479$; ALT: $t = 1.81$, d.f. = 31, $p = 0.080$; YC: $t = 0.54$, d.f. = 15, $p = 0.597$). The birds chose the prosocial option more often during the ALT (mean ± s.d.: 61.5 ± 35.8%) compared to the UNI condition (56.7 ± 36.7%; GLMM: $\beta = 0.38$, s.e. = 0.18, $z = 2.01$, $p = 0.037$), whereas no difference in the number of prosocial choices emerged between the UNI condition and the ACC condition (55.0 ± 38.5%; GLMM: $\beta = -0.12$, s.e. = 0.21, $z = -0.57$, $p = 0.564$; figure 2). Interestingly, and in contrast with the equal rewards conditions, the birds chose the prosocial token less frequently during the YC condition (55.0 ± 37.0%) compared to the ALT condition when unequal rewards were delivered (GLMM: $\beta = -0.50$, s.e. = 0.22, $z = -2.30$, $p = 0.021$). As with the equal ALT condition, we assessed the contingency between the partner's choice in the previous trial and the actor's subsequent token choice and found a significant contingency between partner's and actor's choices ($\chi^2 = 143.98$, d.f. = 1, $p < 0.001$; table 3).

Across the six 10-trial blocks, the birds did not increase the choice for the prosocial option in the UNI condition (GLMM: $\beta = -0.06$, s.e. = 0.23, $z = -0.26$, $p = 0.796$). In the ALT condition, the birds tended to increase the choice for the prosocial option throughout the test (GLMM: $\beta = 0.20$, s.e. = 0.11, $z = 1.83$, $p = 0.069$). A similar trend was seen in the birds that chose the prosocial token more frequently throughout the course of the YC condition (GLMM: $\beta = 0.66$, s.e. = 0.24, $z = 2.81$, $p = 0.005$).

## 4. Discussion

In line with our predictions, we found that African grey parrots exhibited a prosocial tendency towards affiliative partners in a PCT, although such tendency appeared to vary between individuals. This may

have indicated that some individuals understood the task better than others; however, at the group level, the birds did not seem to distinguish between control conditions, in which they could or could not access rewards delivered to both compartments; hence indicating that most, if not all of them, were having problems in understanding the task contingencies (i.e. the spatial arrangement). Despite this lack of understanding, the birds seemed to reciprocate choices made by their partner in the alternating condition and increased their prosocial behaviour compared to the unilateral condition, in which one member of the pair remained in the donor role until all trials were completed and thus not allowing for any immediate reciprocation. This absence of a spontaneous prosocial preference in the UNI condition could have been caused by a lack of feedback from the recipient, who was powerless to respond as they could not directly affect the actor's choices. Furthermore, the birds' prosocial tendencies were affected by inequity. Contrary to our predictions and previous findings in primates, which reduced their prosocial preferences by unequal reward distribution [19], the parrots made more prosocial choices when unequal rewards were delivered in the unilateral condition, while fewer prosocial choices were made in the alternating condition. By contrast, the capuchin monkey increased their prosocial choices by unequal reward distribution if they had the opportunity to alternate their choices with their partner increasing the payoffs of mutual prosociality [22].

While the parrots exhibited no preference for the prosocial token during the unilateral condition, we found that they increased their prosocial choices in the alternating condition, in which choices could be reciprocated, above chance level. In particular, more prosocial tokens were chosen following a prosocial choice made by their partner, and individuals were more likely to select selfish tokens following selfish choices. This indicates that the parrots, in contrast with the capuchins who did not show any evidence for contingent reciprocity [22], followed a tit-for-tat strategy when the roles alternated on a trial-by-trial basis, possibly involving 'attitudinal reciprocity', which is defined as the mirroring of attitudes across short-term intervals [61]. In our specific PCT context, the parrots might have been copying their partner's prosocial choices because of a positive attitude towards their partner, given its previously made prosocial behaviour towards them. In an attempt to understand the underlying motivation of their reciprocal behaviour, i.e. to find out whether the birds responded blindly to the reward contingencies or indeed to the attitude of the second individual, the yoked control condition was implemented. Here, the partner was not allowed to choose the token, but the choices were made by the experimenter instead, who copied the corresponding responses from the previous alternating session, in order to achieve comparability between the two situations. Thus, the potential to influence the partner's attitude with a reciprocal 'tit-for-tat' strategy was eliminated in the yoked control. Results revealed that the parrots chose the prosocial option equally often in the alternating and yoked control condition. By contrast, the capuchins reduced their prosocial choices in the yoked control condition, suggesting that they responded to their partner's behaviour rather than just the presence of the partner or rewards [22]. Given this indifference, it cannot be ruled out that the birds responded simply to the reward contingencies irrespective of whether the partner had an attitude or not, which would be considered as reinforcement learning. Alternatively, it is possible that the parrots were trying to reciprocate choices with the experimenter, as she was the one seemingly selecting the tokens (although she was only copying the choices from the partner in the previous ALT condition).

Contradicting our prediction, an unequal reward distribution did not decrease prosociality. If the birds were sensitive to reward quality, we would have expected them to decrease their prosocial choices in the unilateral condition, while increasing their prosocial behaviour in the alternating condition since choices could be reciprocated (as in [22]). The parrots displayed the exact opposite pattern and significantly increased their prosocial choices in the unilateral condition, but not in the alternating condition, although not exceeding chance level in either condition. The decrease in prosocial choices in the unequal ALT compared to the equal ALT condition may be explained by a frustration effect. The switch from receiving seeds (when in the actor role) after having received walnut in the previous trial (if the partner made a prosocial choice) might be perceived as frustrating because the reward quality had decreased. Potentially, such induced frustration might affect the parrots' other-regarding preferences. Interestingly, the contingency between choices was significant and birds chose the selfish option more often after selfish choices, and prosocial choices were paid back with prosocial ones. This effect was even stronger in the unequal conditions compared to the equal conditions. Potentially, the sight of a walnut might be rewarding enough for the parrots to preferably select the prosocial token, as in the unequal unilateral condition, even though it is delivered to the partner. The increase in prosocial choices across sessions in the unequal alternating condition could additionally be explained by reinforcement learning (i.e. pick same token as partner) or attitudinal reciprocity. Similarly, moderate inequity in a PCT did not lead to an immediate break-

down of prosocial behaviours in capuchin monkeys [55]. Thus, prosocial behaviours can be maintained over a short time despite unequal payoffs. In a previous study, the same birds tested in the present study did not show a negative reaction to inequity if the experimenter rewarded two neighbouring birds differentially for the same task [58]. However, a lack of inequity aversion does not indicate that the parrots were not caring about which reward their partner received, it rather shows that they did not refuse to cooperate in an event of unequal reward distribution. The fact that the birds tolerated minor inequity may even increase the likelihood of finding prosocial attitudes as it shows that the birds are very tolerant of seeing their partner receive something better than themselves. In fact, Brosnan *et al.* [55] found that low levels of inequity can be tolerated while still making prosocial choices. Consequently, if the birds did not show a reaction to unequal treatment and thus seem to be very tolerant to an inequality in the reward distribution, they should be even less restrained to provide food to a conspecific in unequal situations compared to species that are sensitivity to inequity.

Despite exhibiting prosocial tendencies in some of the test conditions, the parrots clearly had problems in understanding the task contingencies as indicated by the behaviour in the control conditions. In particular, we found that contrary to the predictions, they did not switch to a 'prosocial' strategy during the accessible condition, in which they could have maximized their outcome by retrieving the reward from the partner's compartment, while they developed a prosocial tendency in the inaccessible control, although seeing the inaccessible food displayed next to them must have been frustrating. Other species, in which the same control conditions had been implemented to examine whether prosocial choices were made with the goal to deliver food to the neighbouring chamber, obviously understood the test contingencies and responded according to the predictions. Showing the exact opposite pattern to our parrots, capuchin monkeys [22], dogs [62], chimpanzees [20] and other primate species [63] decreased their prosocial choices in the absence of a partner but increased prosocial choices when they could access both compartments. We propose two non-exclusive explanations for the parrots' behaviour in the control conditions: first, the parrots might not have understood the spatial arrangement of the test room. Even though the parrots were habituated to the possibility of accessing the partner's compartment, they might not have been aware that the second reward could be obtained. Still, most birds (with the exception of one bird) consumed both rewards in the case of a prosocial choice in the accessible control, but the delay between choice and consumption of the second reward might have been too long for understanding the contingencies. Furthermore, they might have generalized from the previous test conditions (all birds had been tested in unilateral and alternating conditions before the control conditions) that the partner's compartment cannot be accessed and stuck with this experience instead of exploring the new possibility of crossing to the other side. Interestingly, the birds seemed to learn aspects of the task's contingencies across sessions since they developed a preference for the selfish tokens in the inaccessible control but not in the accessible control. Using a more intuitive set-up (e.g. open barrier partially, as in most other prosocial choice studies [22,62]) might have helped the parrots to succeed (i.e. choose most beneficial option consistently). Alternatively, as a second explanation, one could argue that the parrots were simply indifferent to switching back to the selfish tokens and avoiding the prosocial tokens, which they had favoured in the previous test conditions. Indeed, there is no cost to continue selecting the prosocial tokens in the control conditions, apart from the birds seeing a reward being delivered to the other empty enclosure, something that they might be rather familiar with from other experimental procedures. Likewise, the birds' token choices did not change depending on whether the partner could access the reward (unilateral condition) or not (social facilitation control). This raises the question of why animals should switch their preferences in the control conditions if they were successful with their strategy in previous sessions and if there is no cost in just persisting in it. It is possible that we would have obtained very different results, had there been a notable difference between rewards. For example, if we would have run the control conditions with an unequal reward distribution as well, i.e. using walnut as a more valuable reward in the empty compartment/on the partner's side), the subjects' attention towards the payoff might have been increased.

So far, mixed evidence of prosociality has been reported in birds. In particular, corvids' prosocial tendencies have been in the focus of recent research, revealing that ravens seem to act rather selfishly in PCTs [29,38,64] with controls indicating that they clearly understood the task. Importantly, these studies purposefully excluded the possibility for reciprocity (i.e. keeping fixed roles throughout the experiment), which might have prevented a capacity for prosociality to emerge. On the contrary, other corvid species seem to exhibit prosocial behaviours [35–37]. Interestingly, the studies on pinyon jays and azure-winged magpies allowed reciprocity to arise, since they did not have fixed roles of actor

and partner. Instead, the birds were allowed a completely unrestricted choice, in the case of the group service paradigm [35], or the birds switched roles following each session [37]. Accordingly, the possibility for reciprocity might facilitate the emergence of other-regarding behaviours, maybe because these scenarios are perceived as more natural than paradigms with a restricted partner choice. The only study that assessed parrots' prosocial behaviours so far, found—like in the current study—mixed results in African grey parrots, although those two birds made choices alternately [53]. Indeed, PCTs are highly standardized but therefore arguably rather artificial situations and they are cognitively demanding since the individuals need to keep track of token values, the spatial arrangement of the set-up and the actions of their partners [25]. Accordingly, it might be the case that this set-up exceeded the parrots' cognitive abilities.

Several limitations need to be considered when interpreting the results of the current study. Firstly, the parrots were tested in a fixed dyad composition. Removing the possibility for free partner choices in social/cooperative interactions renders the test situation less natural and ecologically relevant (see [65] for a review), albeit the dyads were composed of affiliated birds. Another consideration to make is that affiliated but not mated partners were tested, which, with 4 years of age, had not yet reached full maturity [42]. Considering that most parrots are monogamous [66], it would be worthwhile investigating if mated pairs show more prosocial behaviours towards each other than unmated affiliative partners and how relationships may affect the birds' response in a PCT paradigm. Indeed, prosocial behaviours differed strongly across dyads. Two dyads acted prosocially, while one dyad exhibited rather selfish behaviour across conditions and the other dyad exhibited no clear selfish or prosocial preferences (see electronic supplementary material, figure S1 and table S3 for individual data). Given our limited sample size, we were not able to further test which factors (e.g. age, sex, relatedness, dominance) drove this variation. Moreover, it might be worthwhile to test parrots in a simplified set-up, giving them only the choice between rewarding the partner or refraining from it, while the actors themselves never get rewarded (0/0 versus 0/1) [16,28]. Here, Burkart & Rueth's [67] study with children, and House et al.'s [68] with chimpanzees found that subjects made more prosocial choices in such 0/0 versus 0/1 scenarios compared to cognitively more demanding 1/1 versus 1/0 rewarding schemes. Given that this approach has successfully been used not only with primates [16] and dogs [28] but also rats [69], it may be a fruitful avenue for cross-species comparisons including non-mammal species.

Social tolerance has been proposed as a proximate mechanism for prosociality, as only individuals that share resources might help conspecifics to obtain a resource as well [13]. Supporting this hypothesis, we found that African grey parrots, a highly social species, do show prosocial tendencies and reciprocate choices in a PCT. The underlying motivation for this reciprocal behaviour, however, remains elusive, since an indifference to deliver food to the empty neighbour compartment and/or a lack of understanding for the spatial contingencies of the set-up, seemed to account for their seemingly arbitrary behaviour in the control conditions. It is the overarching dilemma of animal cognition studies that we do not know how the test animals under investigation perceive a test situation. In particular in comparative studies, experimental procedures should be kept as similar as possible, while still acknowledging species-specific ecology and behaviour. Consequently, the PCT might be well suited for primates [22] and potentially dogs ([62], although see the discrepancy in results to [28]), but might be less ecologically valid for birds. In order to further extend our knowledge about the phylogenetic distribution of other-regarding behaviours, more investigations of prosocial tendencies within the parrot order (*Psittaciformes*) are required, involving both paradigms that have been used on non-avian taxa and more intuitive and ecologically relevant paradigms (e.g. testing dyads of mated individuals) that are specifically designed for parrots (and other monogamous birds like corvids).

Ethics. All applicable international, national and institutional guidelines for the care and use of animals were followed. In accordance with the German Animal Welfare Act of 25 May 1998, Section V, Article 7 and the Spanish Animal Welfare Act 32/2007 of 7 November 2007, Preliminary Title, Article 3, the study was classified as non-animal experiment and did not require any approval from a relevant body.
Data accessibility. Data are available in the electronic supplementary material.
Authors' contributions. A.K. conceived the experiment and drafted the manuscript; D.B. analysed the data and drafted the manuscript; S.B. conceived and carried out the experiment; A.M.P.v.B. conceived the experiment and helped draft the manuscript. All authors gave final approval for publication.
Competing interests. The authors declare no competing interests.
Funding. The study was funded by the Max-Planck Society and a donation by Mrs and Mr Klinger.
Acknowledgements. We thank the Loro Parque and its president, Mr Wolfgang Kiessling for their generous support, the access to the birds and the research facilities. We thank the Loro Parque Fundación and its president Mr Christoph

Kiessling for their collaboration and the staff of the Loro Parque Fundación, the animal caretakers and the veterinary department for their support. We also thank Laurie O'Neill for helpful feedback at the earlier stage of the manuscript.

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
