## [Reviewer comments · Royal Society Open Science]

Review History

RSOS-190696.R0 (Original submission)

Review form: Reviewer 1

Is the manuscript scientifically sound in its present form?

Yes

Are the interpretations and conclusions justified by the results?

Yes

Is the language acceptable?

No

Is it clear how to access all supporting data?

No

Do you have any ethical concerns with this paper?

No

Have you any concerns about statistical analyses in this paper?

No

Recommendation?

Major revision is needed (please make suggestions in comments)

Comments to the Author(s)

The manuscript entitled “Assessing African grey parrots’ prosocial choice tendencies in a token choice paradigm” presents findings from a prosocial choice task in African grey parrots. The study is a thorough replication of previous work done in primates and as such, provides a nice point for comparison in the literature. Overall the methodology was well thought out, the analyses appropriate, the literature review thorough, and the conclusions were sound. I have one major concern with the methods, but all other suggestions or questions are minor, denoted by line numbers below.

Introduction

Line 58: This seems like a bit of an exaggeration, given that there are ample studies that show cooperative behavior among non-kin in primates in the absence of reciprocation. For example, in studies [20,25] there was no opportunity for reciprocity within the experiment and the primates still were cooperative. I suspect the authors are suggesting that nonkin cooperate only when they are familiar (which has been difficult to study since it is challenging to find species and settings where strangers can be tested together). But this line seems like a bit of an overstatement as it is written, especially since in [20] both kin and non-kin pairs behaved similarly.

Line 108: While the parrots succeeded at the string-pulling task in [49], they did not show awareness that a partner was needed in the delay condition. By most accounts, this would be considered a failure at cooperation, since they didn’t appear to notice the role of the partner.

Methods

General comment: I am sure this was unintentional, but at times the language follows Suchak & de Waal [ref 21] extremely closely, even verbatim. I recognize that when doing a replication it may be hard to think of alternative ways to phrase things, but the authors need to extensively re-word large portions of the methods section to avoid violating copyright. For this reason I am suggesting a major revision.

Line 213: Why were the compartment sides fixed, rather than counterbalanced across days/sessions? Could this possibly have inhibited them from moving from side to side in the accessible condition?

Line 220: I think this should say the partner could witness the actor’s behavior

Results:

Line 421: One possibility for this, which hasn’t really been addressed in the discussion is that during the unequal ALT condition, the birds might experience a frustration effect, since in one trial they might get a high value reward as the partner, but in the next get a low value reward as the actor, even though the token chosen in both trials is the same.

Line 443: Are the descriptives the $M \pm SD$? If so, can you specify that?

Line 450: Have you tried a logistic regression using previous choice as a predictor, as opposed to the contingency table? Given that there were differences among individuals/pairs (as discussed

in lines 613-616), it seems possible that certain individuals might be driving this effect, which would be masked in pooled data.

Line 457: I think the interpretation “indicating that the presence of a partner...” does not match the statistic presented here, since in both UNI and SFC the partner was present. It seems to me that this suggests that they weren’t making their choice based on whether or not it was actually helping the partner.

Line 464: It seems odd to me that the INACC condition is the only one in which the birds changed over the 10-trial blocks. I note that half of the birds started with ACC and the other started with INACC. Is it possible this effect (and the overall high number of prosocial choices) is driven by the birds that started with ACC, since in the previous condition choosing the prosocial token was advantageous? Perhaps they needed to learn that they could not access both rewards?

Discussion:

Line 546: I think there is a sentence structure issue in the sentence starting “While their behaviour...”

Line 577: I am not necessarily recommending the authors do this (as I understand how difficult it is to run more conditions), but retesting the birds in the partner absent conditions with fresh tokens could identify whether it was the tokens or the set up causing this issue. If they still stayed to one side with fresh tokens, you would know it is the latter explanation, but if they showed more flexibility with fresh tokens, it would likely indicate that they formed fixed preferences for the tokens used in previous conditions.

General comment: it seems to me that the two PCT studies [the current study and ref. 53] combined with the string pulling task [ref 49], all suggest that the parrots may not be monitoring and responding to their partner’s behavior in the same way that we see in other species. This is not to suggest that they cannot truly cooperate by taking into account the partner’s behavior [c.f. Ref 64], but perhaps tasks originally designed for primates are not tapping into these abilities in a parrot-friendly way. This is not meant as a criticism of this particular study (how else would we know except to try tests that have already been designed?) but rather something to consider adding to the discussion as it provides a suggestion for people in the field to think outside the primate box, so to speak.

Review form: Reviewer 2

Is the manuscript scientifically sound in its present form?

Yes

Are the interpretations and conclusions justified by the results?

No

Is the language acceptable?

Yes

Is it clear how to access all supporting data?

Yes

Do you have any ethical concerns with this paper?

No

Have you any concerns about statistical analyses in this paper?

I do not feel qualified to assess the statistics

Recommendation?

Major revision is needed (please make suggestions in comments)

Comments to the Author(s)

Krashennikova and colleagues have written an extremely interesting paper on prosocial behaviour in Grey parrots. They examine multiple possible situations and test the birds extensively. I nevertheless have several comments – some involving minor clarification, others a bit more substantive – which I present in order of appearance.

L. 67: Would be clearer as: "...the various hypotheses that have been proposed to explain..."

L. 69: Should read: "our understanding not only of..." [in contrast to 'but also']

L. 84: Would be clearer as "Similar to the findings on primates, those on corvid prosociality are mixed....pattern as yet."

L. 86: For the same reason: "...other paradigms, jackdaws (*Corvus monedula*), for example, show..."

L. 94: Should read: "...also making them interesting...candidates in which to investigate..."

L. 100: "...they likely form..."

L. 114: to be accurate, the line should read "...chose among four options that could reward..."

L. 117ff: This description of the material is not correct. Citing from the paper itself: "However, whether they shared with their partner at no cost to themselves seemed, at least for our subjects, to depend upon their dominance status and upon who led and who followed. Griffin has always been dominant to Arthur, and Figs. 3 and S1 show that when Griffin led, he became selfish, but when placed in the position of follower, he acted in a more sharing manner." That is, when Griffin led he started out sharing but as Arthur became more selfish over time, Griffin also became more selfish. When Griffin was the follower, he retained his more sharing manner. Arthur, the subordinate, "...started with a slight giving tendency as both leader and follower, then became statistically significantly more selfish both as leader and follower."

L. 122: Should read "...by decreasing or increasing prosocial choices appropriately". [Must maintain parallel construction – selfish/decrease; generous/increase]

L. 123: This sentence is not correct – the authors tested for side biases.

L. 124: This sentence is also not correct. Control was inherent in the task: Note that the birds did not copy the human, as the birds were not *giving* treats to the generous human but *sharing* treats, and they quickly learned to avoid the null cup (no treats to anyone), demonstrating that they clearly understood the task. Note, too, the follow-up study that further tested the task's assumptions....

L. 126: Should read: "objective of the present study was to more fully investigate..."

L. 131: remove “meant to”

L. 136: Should read “...that would provide...”

L. 140: Would be clearer if read: “..(2) this tendency may be...”

L. 173: exchange tokens?

L. 206: Could the inequity aversion study have affected these results?

L. 209: I'm confused....There are three clutches, and all birds are the same age...How many clutches/yr by the same parents? Lizzy and Nina were same-sexed siblings from the same clutch and Nikki and Jack were opposite-sexed siblings from the same clutch; Then Kimmi and Jelo were siblings but from different clutches? Ditto for Bella and Sensei? If the birds are from different clutches by the same parents, one is likely to be older and more dominant...Did the authors test for any type of dominance? In the Péron et al. study, dominance turned out to be important....

L. 214: I'm confused: Counterbalanced with respect to what? If the sides are fixed, what is being counterbalanced? With respect to different experiments?

L. 216: Even if tokens were jumbled, did birds have a position preference, separate from color?

L. 220: I'm confused...do the authors mean “recipient” in at least one of these places where “actor” is used?

L. 236: At one level I understand why the actor was fed first, but on another level, isn't that a form of inequality? Wouldn't it be better to be simultaneous? Or did the authors want the actor to see the recipient's behaviour and vice versa?

L. 240: I certainly understand why a trial would be cancelled in this instance, but how many times did it happen in each condition? Isn't it an indication of frustration? It would seem to be akin to the choice of the purple cup in Péron et al....

L. 245: What is a 'tiny' reward, given that one sunflower seed isn't exactly a large reward...? Thirty seeds at any one time is quite a bonanza, however....Did birds' motivation wane toward the end of sessions? Also, L. 247: Again, totally legitimate, but how many times did it happen?

L. 265: I'm confused...if one session had forced choice trials, what was the other session?

L. 281: So, the actor first gets 30 (or 60?) trials in which it gets a reward no matter what it is forced to do, but the experimenter expects that the actor is actually paying attention to what happens to its partner, so that it presumably learns what to do to give the partner a reward...Might only the partner be attending to what is happening, because the choice affects only the partner? So the first bird to be unilateral doesn't care, but the second bird to be unilateral does care, and does pay attention...What I'm asking is whether there's a difference in prosocial behavior depending on order. If the birds paid a lot of attention to what was happening during the familiarization trials to the partner, it shouldn't matter, but did they really make the association?

L. 300: Do the authors mean “consecutive” rather than “subsequent” ...i.e., one right after another?

L. 301: Actually, the actor seems to have 60 free-choice trials, split into two sessions...yes? [Not clear as written...]

L. 337: I'm confused....why wouldn't the actor then treat the experimenter as the partner, responding to whatever the partner did? Wouldn't you get roughly the same results? The task is testing whether the bird responds to another bird versus the experimenter, not another bird versus the reward. I understand that the experimenter actually provides the rewards in both cases, but if bird A doesn't see bird B making a choice and only sees the experimenter's actions, then there is an obvious difference. Or maybe bird A thinks that its choice makes the experimenter act twice in some way? [That is...the experimenter distributes treats once based on bird A's choice and then once again without anything else happening...] I would think that this is an extremely confusing condition. Maybe the task would work if it was automated in some way, so that there was no visible actor, but in this case...it is confusing.

L. 362ff: Again, how many times did the birds refuse to enter the empty compartment? Makes a difference in their understanding if they chose prosocially but didn't access the additional treat... And it might make a difference as to which birds started with the inaccessible versus which started with accessible conditions... If the birds started with inaccessible, and saw the experimenter remove the food after 2-3 sec, maybe they assumed it was forbidden and figured that they should not try to access it?

L. 386: What did the birds understand about the blockage? Was this an alternating condition or not? If not alternating, maybe only the bird that didn't get the treat understood the condition and would respond appropriately? Maybe bird A keeps being prosocial because it wants bird B to see that it is trying to be generous no matter what the experimenter does?

L. 407ff: The birds that exhibited preferences – were they first actors or first recipients?

L. 411: I think the authors mean S2...

L. 437: Again...did the first actor understand the issues? Only when the first recipient became the actor would the task make sense...and if the first actor wasn't very prosocial, the recipient might not be so after it became the actor when there were 30 trials of one type involved....

L. 446: Again – possibly the birds were trying to reciprocate with the experimenter in some way, so that it was still some kind of social situation...

L. 457: I'm confused...a partner was present in both UNI and SFC, so why would an equal response suggest that the presence of a partner did not affect their choices? Would have to contrast data between the presence and absence of a partner to make that point...

L. 507: Yes, they were having trouble understanding the task contingencies, but not issues of whether or not a partner was present, absent, or able to access the reward...putting oneself in the situation of the bird, a human might have trouble figuring out the various task contingencies.

L. 511ff: But by the time the unequal reward task was administered, the birds had a pretty good idea that turns would be taken, whether immediately or eventually...so they could reason that if they ever wanted the walnut, they'd better give it to their partner...and the effect seems driven mostly by one dyad...

L. 530: Theoretically the option of tit-for-tat was eliminated, but one would have to show that the synergy was different to make such a claim. It was still a social system in which the birds could have been trying to influence the experimenter...

L. 546: Remove comma...sentence unclear....

L. 605: Maybe the problem is not about the birds' cognitive abilities – the experimenters did not examine social dominance issues or the extent of the 'bonding' within the pairs. Dyad 2 here seemed particularly uncooperative; I wonder if they were the least strongly bonded pair or the one with the most difference in dominance...I simply think that the tasks were confusing, and that in some instances other issues such as dominance and the order in which a bird in a pair began a task may have affected the outcomes.

We also do not know how many times the birds simply refused to work...Such behaviour is common in parrots and is not a reflection on the study, but knowing the numbers of failed trials may provide a handle on how confusing the tasks may have been.

I'm very conflicted about this paper – the experimenters tried to design appropriate controls but mainly seemed to have confused their subjects. We know nothing about the number of failed trials and nothing about the actual relationships between the birds in the dyads. The authors are honest in their overall evaluation that additional work is necessary to determine prosociality in parrots (at least Grey parrots), but leave us without much more information than we had from the earlier studies (i.e., that parrots like to play tit-for-tat to some extent, although the previous studies demonstrated that the parrots didn't simply copy the behavior of their partner – at least when the partner was human).

Decision letter (RSOS-190696.R0)

03-Jul-2019

Dear Dr Krasheninnikova,

The editors assigned to your paper ("Assessing African grey parrots' prosocial tendencies in a token choice paradigm") have now received comments from reviewers. We would like you to revise your paper in accordance with the referee and Associate Editor suggestions which can be found below (not including confidential reports to the Editor). Please note this decision does not guarantee eventual acceptance.

Please submit a copy of your revised paper before 26-Jul-2019. Please note that the revision deadline will expire at 00.00am on this date. If we do not hear from you within this time then it will be assumed that the paper has been withdrawn. In exceptional circumstances, extensions may be possible if agreed with the Editorial Office in advance. We do not allow multiple rounds of revision so we urge you to make every effort to fully address all of the comments at this stage. If deemed necessary by the Editors, your manuscript will be sent back to one or more of the original reviewers for assessment. If the original reviewers are not available, we may invite new reviewers.

- Data accessibility

If you wish to submit your supporting data or code to Dryad (<http://datadryad.org/>), or modify your current submission to dryad, please use the following link:
<http://datadryad.org/submit?journalID=RSOS&manu=RSOS-190696>

- Competing interests

- Authors' contributions

- Acknowledgements

- Funding statement

Kind regards,

Alice Power

Editorial Coordinator

on behalf of Kevin Padian (Subject Editor)

Associate Editor's comments:

Two reviewers have commented on your report. Both raise a number of concerns regarding, for instance, the methodology employed, the literature review, and how closely your text matches previously published work. To comment on the latter point, the journal's editors recognise a degree of overlap in methods may be inevitable, but would nevertheless strongly encourage the authors to seek alternative phrasing and to ensure they fully reference methodologies borrowed from elsewhere to ensure that A) your work is as original a contribution as practical and B) earlier work receives the credit it is due. A more general comment on the paper is that it is clear a substantial effort is required to get the paper to publication. You will need to work hard to persuade both the editors and reviewers that the revised manuscript has improved sufficiently to consider publication: you will need to include a point-by-point response detailing changes or any rebuttals, and also - ideally - include a tracked changes version of the manuscript to make clear how you've effectively responded to the critiques. Good luck, and we'll look forward to receiving the revision - the original reviewers will be invited to look at your responses, so do make sure they are clear and effective.

Subject Editor Comments to Authors:

Please take into account all the comments of the AE and reviewers. If you need more time than allotted for the revision, do let the office know and it will be no problem. Best wishes.

Comments to Author:

Reviewers' Comments to Author:

Reviewer: 1

Comments to the Author(s)

The manuscript entitled "Assessing African grey parrots' prosocial choice tendencies in a token choice paradigm" presents findings from a prosocial choice task in African grey parrots. The study is a thorough replication of previous work done in primates and as such, provides a nice point for comparison in the literature. Overall the methodology was well thought out, the analyses appropriate, the literature review thorough, and the conclusions were sound. I have one

major concern with the methods, but all other suggestions or questions are minor, denoted by line numbers below.

Introduction

Line 58: This seems like a bit of an exaggeration, given that there are ample studies that show cooperative behavior among non-kin in primates in the absence of reciprocation. For example, in studies [20,25] there was no opportunity for reciprocity within the experiment and the primates still were cooperative. I suspect the authors are suggesting that nonkin cooperate only when they are familiar (which has been difficult to study since it is challenging to find species and settings where strangers can be tested together). But this line seems like a bit of an overstatement as it is written, especially since in [20] both kin and non-kin pairs behaved similarly.

Line 108: While the parrots succeeded at the string-pulling task in [49], they did not show awareness that a partner was needed in the delay condition. By most accounts, this would be considered a failure at cooperation, since they didn't appear to notice the role of the partner.

Methods

General comment: I am sure this was unintentional, but at times the language follows Suchak & de Waal [ref 21] extremely closely, even verbatim. I recognize that when doing a replication it may be hard to think of alternative ways to phrase things, but the authors need to extensively reword large portions of the methods section to avoid violating copyright. For this reason I am suggesting a major revision.

Line 213: Why were the compartment sides fixed, rather than counterbalanced across days/sessions? Could this possibly have inhibited them from moving from side to side in the accessible condition?

Line 220: I think this should say the partner could witness the actor's behavior

Results:

Line 421: One possibility for this, which hasn't really been addressed in the discussion is that during the unequal ALT condition, the birds might experience a frustration effect, since in one trial they might get a high value reward as the partner, but in the next get a low value reward as the actor, even though the token chosen in both trials is the same.

Line 443: Are the descriptives the $M \pm SD$? If so, can you specify that?

Line 450: Have you tried a logistic regression using previous choice as a predictor, as opposed to the contingency table? Given that there were differences among individuals/pairs (as discussed in lines 613-616), it seems possible that certain individuals might be driving this effect, which would be masked in pooled data.

Line 457: I think the interpretation "indicating that the presence of a partner..." does not match the statistic presented here, since in both UNI and SFC the partner was present. It seems to me that this suggests that they weren't making their choice based on whether or not it was actually helping the partner.

Line 464: It seems odd to me that the INACC condition is the only one in which the birds changed over the 10-trial blocks. I note that half of the birds started with ACC and the other started with INACC. Is it possible this effect (and the overall high number of prosocial choices) is driven by the birds that started with ACC, since in the previous condition choosing the prosocial token was advantageous? Perhaps they needed to learn that they could not access both rewards?

Discussion:

Line 546: I think there is a sentence structure issue in the sentence starting “While their behaviour...”

Line 577: I am not necessarily recommending the authors do this (as I understand how difficult it is to run more conditions), but retesting the birds in the partner absent conditions with fresh tokens could identify whether it was the tokens or the set up causing this issue. If they still stayed to one side with fresh tokens, you would know it is the latter explanation, but if they showed more flexibility with fresh tokens, it would likely indicate that they formed fixed preferences for the tokens used in previous conditions.

General comment: it seems to me that the two PCT studies [the current study and ref. 53] combined with the string pulling task [ref 49], all suggest that the parrots may not be monitoring and responding to their partner’s behavior in the same way that we see in other species. This is not to suggest that they cannot truly cooperate by taking into account the partner’s behavior [c.f. Ref 64], but perhaps tasks originally designed for primates are not tapping into these abilities in a parrot-friendly way. This is not meant as a criticism of this particular study (how else would we know except to try tests that have already been designed?) but rather something to consider adding to the discussion as it provides a suggestion for people in the field to think outside the primate box, so to speak.

Reviewer: 2

Comments to the Author(s)

Krashennikova and colleagues have written an extremely interesting paper on prosocial behaviour in Grey parrots. They examine multiple possible situations and test the birds extensively. I nevertheless have several comments – some involving minor clarification, others a bit more substantive – which I present in order of appearance.

L. 67: Would be clearer as: “...the various hypotheses that have been proposed to explain...”

L. 69: Should read : “our understanding not only of...” [in contrast to ‘but also’]

L. 84: Would be clearer as “Similar to the findings on primates, those on corvid prosociality are mixed....pattern as yet.”

L. 86: For the same reason: “...other paradigms, jackdaws (*Corvus monedula*), for example, show...”

L. 94: Should read: “...also making them interesting...candidates in which to investigate...”

L. 100: “...they likely form...”

L. 114: to be accurate, the line should read “...chose among four options that could reward...”

L. 117ff: This description of the material is not correct. Citing from the paper itself: “However, whether they shared with their partner at no cost to themselves seemed, at least for our subjects, to depend upon their dominance status and upon who led and who followed. Griffin has always been dominant to Arthur, and Figs. 3 and S1 show that when Griffin led, he became selfish, but when placed in the position of follower, he acted in a more sharing manner.” That is, when

Griffin led he started out sharing but as Arthur became more selfish over time, Griffin also became more selfish. When Griffin was the follower, he retained his more sharing manner. Arthur, the subordinate, "...started with a slight giving tendency as both leader and follower, then became statistically significantly more selfish both as leader and follower."

L. 122: Should read "...by decreasing or increasing prosocial choices appropriately". [Must maintain parallel construction – selfish/decrease; generous/increase]

L. 123: This sentence is not correct – the authors tested for side biases.

L. 124: This sentence is also not correct. Control was inherent in the task: Note that the birds did not copy the human, as the birds were not *giving* treats to the generous human but *sharing* treats, and they quickly learned to avoid the null cup (no treats to anyone), demonstrating that they clearly understood the task. Note, too, the follow-up study that further tested the task's assumptions....

L. 126: Should read: "objective of the present study was to more fully investigate..."

L. 131: remove "meant to"

L. 136: Should read "...that would provide..."

L. 140: Would be clearer if read: "...(2) this tendency may be..."

L. 173: exchange tokens?

L. 206: Could the inequity aversion study have affected these results?

L. 209: I'm confused....There are three clutches, and all birds are the same age...How many clutches/yr by the same parents? Lizzy and Nina were same-sexed siblings from the same clutch and Nikki and Jack were opposite-sexed siblings from the same clutch; Then Kimmi and Jelo were siblings but from different clutches? Ditto for Bella and Sensei? If the birds are from different clutches by the same parents, one is likely to be older and more dominant...Did the authors test for any type of dominance? In the Péron et al. study, dominance turned out to be important....

L. 214: I'm confused: Counterbalanced with respect to what? If the sides are fixed, what is being counterbalanced? With respect to different experiments?

L. 216: Even if tokens were jumbled, did birds have a position preference, separate from color?

L. 220: I'm confused...do the authors mean "recipient" in at least one of these places where "actor" is used?

L. 236: At one level I understand why the actor was fed first, but on another level, isn't that a form of inequality? Wouldn't it be better to be simultaneous? Or did the authors want the actor to see the recipient's behaviour and vice versa?

L. 240: I certainly understand why a trial would be cancelled in this instance, but how many times did it happen in each condition? Isn't it an indication of frustration? It would seem to be akin to the choice of the purple cup in Péron et al....

L. 245: What is a 'tiny' reward, given that one sunflower seed isn't exactly a large reward...?

Thirty seeds at any one time is quite a bonanza, however....Did birds' motivation wane toward the end of sessions? Also, L. 247: Again, totally legitimate, but how many times did it happen?

L. 265: I'm confused...if one session had forced choice trials, what was the other session?

L. 281: So, the actor first gets 30 (or 60?) trials in which it gets a reward no matter what it is forced to do, but the experimenter expects that the actor is actually paying attention to what happens to its partner, so that it presumably learns what to do to give the partner a reward...Might only the partner be attending to what is happening, because the choice affects only the partner? So the first bird to be unilateral doesn't care, but the second bird to be unilateral does care, and does pay attention...What I'm asking is whether there's a difference in prosocial behavior depending on order. If the birds paid a lot of attention to what was happening during the familiarization trials to the partner, it shouldn't matter, but did they really make the association?

L. 300: Do the authors mean "consecutive" rather than "subsequent" ...i.e., one right after another?

L. 301: Actually, the actor seems to have 60 free-choice trials, split into two sessions...yes? [Not clear as written...]

L. 337: I'm confused....why wouldn't the actor then treat the experimenter as the partner, responding to whatever the partner did? Wouldn't you get roughly the same results? The task is testing whether the bird responds to another bird versus the experimenter, not another bird versus the reward. I understand that the experimenter actually provides the rewards in both cases, but if bird A doesn't see bird B making a choice and only sees the experimenter's actions, then there is an obvious difference. Or maybe bird A thinks that its choice makes the experimenter act twice in some way? [That is...the experimenter distributes treats once based on bird A's choice and then once again without anything else happening...] I would think that this is an extremely confusing condition. Maybe the task would work if it was automated in some way, so that there was no visible actor, but in this case...it is confusing.

L. 362ff: Again, how many times did the birds refuse to enter the empty compartment? Makes a difference in their understanding if they chose prosocially but didn't access the additional treat... And it might make a difference as to which birds started with the inaccessible versus which started with accessible conditions... If the birds started with inaccessible, and saw the experimenter remove the food after 2-3 sec, maybe they assumed it was forbidden and figured that they should not try to access it?

L. 386: What did the birds understand about the blockage? Was this an alternating condition or not? If not alternating, maybe only the bird that didn't get the treat understood the condition and would respond appropriately? Maybe bird A keeps being prosocial because it wants bird B to see that it is trying to be generous no matter what the experimenter does?

L. 407ff: The birds that exhibited preferences – were they first actors or first recipients?

L. 411: I think the authors mean S2...

L. 437: Again...did the first actor understand the issues? Only when the first recipient became the actor would the task make sense...and if the first actor wasn't very prosocial, the recipient might not be so after it became the actor when there were 30 trials of one type involved....

L. 446: Again – possibly the birds were trying to reciprocate with the experimenter in some way, so that it was still some kind of social situation...

L. 457: I'm confused...a partner was present in both UNI and SFC, so why would an equal response suggest that the presence of a partner did not affect their choices? Would have to contrast data between the presence and absence of a partner to make that point...

L. 507: Yes, they were having trouble understanding the task contingencies, but not issues of whether or not a partner was present, absent, or able to access the reward...putting oneself in the situation of the bird, a human might have trouble figuring out the various task contingencies.

L. 511ff: But by the time the unequal reward task was administered, the birds had a pretty good idea that turns would be taken, whether immediately or eventually...so they could reason that if they ever wanted the walnut, they'd better give it to their partner...and the effect seems driven mostly by one dyad...

L. 530: Theoretically the option of tit-for-tat was eliminated, but one would have to show that the synergy was different to make such a claim. It was still a social system in which the birds could have been trying to influence the experimenter...

L. 546: Remove comma...sentence unclear....

L. 605: Maybe the problem is not about the birds' cognitive abilities – the experimenters did not examine social dominance issues or the extent of the 'bonding' within the pairs. Dyad 2 here seemed particularly uncooperative; I wonder if they were the least strongly bonded pair or the one with the most difference in dominance...I simply think that the tasks were confusing, and that in some instances other issues such as dominance and the order in which a bird in a pair began a task may have affected the outcomes.

We also do not know how many times the birds simply refused to work...Such behaviour is common in parrots and is not a reflection on the study, but knowing the numbers of failed trials may provide a handle on how confusing the tasks may have been.

I'm very conflicted about this paper – the experimenters tried to design appropriate controls but mainly seemed to have confused their subjects. We know nothing about the number of failed trials and nothing about the actual relationships between the birds in the dyads. The authors are honest in their overall evaluation that additional work is necessary to determine prosociality in parrots (at least Grey parrots), but leave us without much more information than we had from the earlier studies (i.e., that parrots like to play tit-for-tat to some extent, although the previous studies demonstrated that the parrots didn't simply copy the behavior of their partner – at least when the partner was human).

Author's Response to Decision Letter for (RSOS-190696.R0)

See Appendix A.

RSOS-190696.R1 (Revision)

Review form: Reviewer 1

Is the manuscript scientifically sound in its present form?

Yes

Are the interpretations and conclusions justified by the results?

Yes

Is the language acceptable?

Yes

Do you have any ethical concerns with this paper?

No

Have you any concerns about statistical analyses in this paper?

No

Recommendation?

Accept as is

Comments to the Author(s)

I sincerely thank the authors for their thoughtful responses to my earlier concerns. At this point I have reviewed the revised manuscripts and have no further questions or concerns to raise.

Review form: Reviewer 2

Is the manuscript scientifically sound in its present form?

No

Are the interpretations and conclusions justified by the results?

No

Is the language acceptable?

Yes

Do you have any ethical concerns with this paper?

No

Have you any concerns about statistical analyses in this paper?

No

Recommendation?

Major revision is needed (please make suggestions in comments)

Comments to the Author(s)

Krashennikova and colleagues have revised their extremely interesting paper on prosocial behaviour in Grey parrots. They examine multiple possible situations and test the birds

extensively. I nevertheless still have several comments – some involving minor clarification, others a bit more substantive – which I present in order of appearance.

L. 29: I'd be more comfortable with something like “..their lack of understanding of the contingencies of the particular tasks used in this study, the underlying...” [that is the most that the authors can claim]

L. 39: This paper is on pro-sociality, which the authors just described as being different from altruism. So where does this sentence with respect to altruism fit? Do they mean pro-sociality? Or at least reciprocal altruism?

L. 58-59: The edit is odd...leave out comma after “familiar”...and maybe “and in situations without a possibility...”

L. 118: Remove comma after “While”...Based on the material in the paper, maybe the description should read “did not show clear reciprocity...” Again, read my comments based on the material in the paper cited: When the dominant bird led, he STARTED OUT prosocial but switched to selfish when his partner consistently acted in a selfish manner”...what is written in this manuscript is not correct.

This description of the material is not correct. Citing from the paper itself: “However, whether they shared with their partner at no cost to themselves seemed, at least for our subjects, to depend upon their dominance status and upon who led and who followed. Griffin has always been dominant to Arthur, and Figs. 3 and S1 show that when Griffin led, he became selfish, but when placed in the position of follower, he acted in a more sharing manner.” That is, when Griffin led he started out sharing but when Arthur became more selfish, Griffin became more selfish. When Griffin followed, he retained his more sharing manner. Arthur, the subordinate, “..started with a slight giving tendency as both leader and follower, then became statistically significantly more selfish both as leader and follower.”

L. 120ff: THEREFORE, these lines should read: “If the dominant bird started the session choosing first, he initially chose the prosocial token, but became more selfish after he experienced selfish behavior from his partner. When the dominant bird was the follower, he retained his prosociality. The subordinate bird, however, became more selfish over time in both leader and follower roles. Therefore, both birds developed...”

L. 125: This sentence is not correct – and the sentence has not been altered....I repeat: the authors tested for side biases and the current authors cannot disregard these tests. I will change my decision to “reject” rather than revise if this revision is not made in a subsequent version.

L. 126: This sentence is also not correct. Control was inherent in the task: Note that the birds did not copy the human, as the birds were not giving treats to the generous human but sharing treats, and they quickly learned to avoid the null cup (no treats to anyone), demonstrating that they clearly understood the task. Note, too, the follow-up study that further tested the task's assumptions. The authors of the current paper cannot blithely disregard data and criticize another paper without full justification. Avoiding the null cup if it gives no reward can be described any way you choose, but it still involves understanding the contingencies of the task. And explaining the choice of the option that provides two rewards rather than one as reinforcement learning is meaningless unless the chooser GETS more rewards. If the chooser gets to share, that means the chooser understands the contingencies of the choice. As for the control without a partner – the subject is simply likely going to be confused about what is going on – see more on that later.

The argument about color bias is justified, however.

L. 134: I think the authors mean “employed” rather than “applied”

L. 201: “without shells”

L. 199: This sentence is not clearly written. Try dividing the sentence into two. “...in all equal reward conditions. When the actor chose prosocially in the unequal condition, in contrast, the partner received a piece of walnut (preference for walnut was assessed prior to the experiment) while the actor received merely the same sunflower seed as in the equal condition.”

L. 217: I think it is important to note here that these birds did NOT express inequity aversion previously. Such finding suggest that the authors already know that the birds are not that likely to be affected by their partner getting a better reward, and that this is a further test of that finding.

L. 216: In the response to reviewers, the authors mention that they did not find any position preference in token choice...this information should be provided somewhere in the results section.

L. 226: Is this a new paragraph?

L. 244: Sentence a bit confusing. Divide sentences. “The rewards inside the cups were not at that time visible to the birds. Once a choice was made, the experimenter lifted the lid(s) off the small food container(s), tilted the cup(s) forward so that both birds could see the reward(s) inside, and then distributed the reward(s).”

L. 259ff: If birds don’t care about inequality, why would they care if one got fed and one didn’t? Isn’t that finding a problem for this entire paper? The receiver could eventually associate the ring color with whether it gets food, but not necessarily with the actions of its partner if it doesn’t care that its partner gets fed and it doesn’t, especially if it was going to get at least half the standard reward just for hanging out near the reward site when the experimenter wanted it to move back. If a whole seed is a standard reward, half that reward is hardly “tiny” [and, yes, I do understand that sunflower seeds cannot be divided into much smaller pieces and still easily be handled, but maybe the authors should have used some other, less favored food?]

L. 270: 12 sessions needed to be repeated? I’m confused...there are only 9 sessions listed in Table 1... If this statement is not a typo, then there are a LOT of repetitions. Could the authors please let the reader know how many sessions were repeated for each condition? That would help the reader know which conditions were more problematic. The authors gave some of the information in their response but not in the paper...it seems as though one dyad really had issues, and it would be helpful to know which one. Maybe that set of data needs to be dealt with separately?

L. 273: What happened after the birds chose a token in the color preference test? Did it just get taken away? Did they get something or nothing for making the choice? If they had any experience in trading tokens for treats, and they chose a token and got nothing, they would – if they had any sense – alternate choice of tokens to see if choosing a different token would give a reward, even if they liked their first choice. It seems that most birds started with a color preference and those colors were eliminated, but it also seems possible that the birds just LEARNED at this point that they should choose randomly because nothing would happen almost whatever they did BUT if they chose their favorite it would disappear. I don’t disagree that one has to check for color preferences, but it’s a rather tricky procedure as they learn to ignore any contingencies connected with the color.

L. 286ff: So now a bird was given “x” to choose and got a reward half the time and was given “y” to choose the other half and still got a reward; sometimes their partner got a reward and sometimes the partner did not. Given that they didn’t care about reward inequality, why would they pay much attention to what was happening, particularly if they were actor but even if they were the powerless recipient? If the recipient couldn’t do anything about what happens, and can’t tell the actor what to do, why pay too much attention?

L. 298: And I’m still confused...The authors mention equal and unequal rewards earlier, but Table 1 does not, so I’m confused. Does each bird have one session as actor with equal rewards then one with unequal reward before the roles are reversed (i.e., that’s what is meant by 2 sessions in Table 1)? Or do reversals occur after equal reward and then again before unequal reward? And do all these trials occur before the alternating sessions? I need to understand what is happening at this point in the paper. Or do the birds cycle through UNI, ALT, YOK in the equal reward before doing unequal rewards? [It seems that way, given that different tokens were used, but I really had to work to figure this out and I’m still not sure.] If I were trying to replicate this experiment, I’d never be able to figure out exactly what was done. The material now taken out helped me before...

L. 305: Two sessions EACH? Including equal or unequal rewards?

L. 343: Do the authors mean 30 trials per bird acting as actor and 30 as recipient? And where do the equal/unequal fit in here...one session with equal rewards, one session with unequal rewards? Not clear...

L. 355ff: I absolutely do not understand the authors’ response to my query. Why would switching to a selfish strategy be the expected response? The experimenter is NOT always being selfish, so why would the actor be selfish? The authors are making AN ASSUMPTION that the birds are treating the experimenter as an automaton, and ignoring the possibility that they were playing with the experimenter. In previous studies, didn’t the experimenter played partner-like roles? So why ignore that now? Given that there were lots of prosocial choices in the alternating condition, the experimenter was being prosocial a lot, so why wouldn’t the actor reciprocate to that? I definitely appreciate the new material added around L. 580.

L. 384: Accessible vs inaccessible: So, I’m a bird and you train me to go back and forth between compartments. I learn that I can get food in both places, but not with respect to anything to do with token choice; I just learn that it’s possible. And there’s no bird next door. If I’m Jelo or Bella or Sensi or Nikki (half the birds), I don’t notice whether or not the passage is open but I choose to put food on the other side because I remember that it’s possible to get food. If I’m Kizzi or Nina or Jack, whether or not the passage is open I’m not prosocial maybe because there’s no bird there. If I’m Kimmi, I’m just confused. Actually, if the birds differentiated between the conditions, they should NOT be random but choose selfish in inaccessible because they’d otherwise be wasting food, and they should choose prosocial in the accessible condition to get more food. But none of the birds (except a tendency in Jack, in the opposite to what would be expected) seemed to understand the different conditions at all. They seemed to hit on some strategy simply with respect to the possibility of more food – hope for the best or accept the worst – and stick to it. I don’t see the task as showing much of anything about understanding cooperation. I think the authors’ comments in the Discussion about what they did and didn’t learn and spatial confusion make more sense than the assumption that the task would be useful, particularly given that the birds’ behaviours altered over time.

Re blockage...about half the birds seemed to understand the issue and the other half did not.

Overall: Individual differences seem to be extremely important in this study, from looking at the supplemental data. Maybe some Greys are good buddies and others simply are not, or maybe some birds are dim and others not...the authors really need to look at and discuss the individuals, not just the group behaviour.

L. 451ff: I'm confused...if birds DID develop preferences, and there were so many individual differences in the results, shouldn't the experimenters have checked specifically to see if the birds' actions were affected by these preferences? Just looking at group behaviour is not sufficient, as the individuals varied so strongly.

L. 474: The authors must mean dashed red line, not dotted.....I'm not sure what the dotted line means.

L. 560: Correct English is "fewer prosocial", not "less prosocial"...

We still do not know the relationships between the individuals in the pairs, although the authors do admit to the artificiality of the situation. Note: A 'good' relationship doesn't necessarily mean equal; if the birds have worked out a dominant-subordinate relationship and are happy with it, that makes it "good"...but if there is conflict, that is a different situation, and could easily affect the results.

We also know that parrots engage in courtship feeding, where one bird gives up food to the other with no reciprocity, and in Greys both parents feed the young directly from what they obtain when they forage (unlike mammals, who suckle, which is quite different in terms of sharing resources)...so the whole idea of unequal reward as an issue is quite different from what one would see in primates.

I'm still very conflicted about this paper – given how critical the experimenters are of the previous studies, their failure to examine the material of the previous studies carefully, and their claims that they performed their experiments in order to correct all the problems they perceived in the previous studies, they really have to be exceptionally careful about what they are saying and claiming. I re-read the paper even more carefully this time, and found other inconsistencies that need to be addressed. I'm not trying to be obnoxious – I don't want someone else to take them apart after the paper is published. I think that looking at the behaviour both of individuals and individual pairs would be very helpful.

Again, the good part of this paper is that the authors are honest in their overall evaluation that additional work is necessary to determine prosociality in parrots (at least Grey parrots). And, yes they did test more birds than in previous studies, but looking at the individual data it seems that some birds are good at being prosocial, others are not, and some learn more about the situation than do others. And I repeat that their controls were confusing...if I had been in the birds' place, I would have been confused. Just because someone else used these controls doesn't mean they are correct. Thus, again, the authors have to be extremely careful about their analyses and claims as well as criticisms.

Decision letter (RSOS-190696.R1)

22-Aug-2019

Dear Dr Krasheninnikova:

Manuscript ID RSOS-190696.R1 entitled "Assessing African grey parrots' prosocial tendencies in a token choice paradigm" which you submitted to Royal Society Open Science, has been reviewed. The comments of the reviewer(s) are included at the bottom of this letter.

Please submit a copy of your revised paper before 14-Sep-2019. Please note that the revision deadline will expire at 00.00am on this date. If we do not hear from you within this time then it will be assumed that the paper has been withdrawn. In exceptional circumstances, extensions may be possible if agreed with the Editorial Office in advance. We do not allow multiple rounds of revision so we urge you to make every effort to fully address all of the comments at this stage. If deemed necessary by the Editors, your manuscript will be sent back to one or more of the original reviewers for assessment. If the original reviewers are not available we may invite new reviewers.

- Ethics statement

- Data accessibility

- Competing interests

- Authors' contributions

All submissions, other than those with a single author, must include an Authors' Contributions section which individually lists the specific contribution of each author. The list of Authors

should meet all of the following criteria; 1) substantial contributions to conception and design, or acquisition of data, or analysis and interpretation of data; 2) drafting the article or revising it critically for important intellectual content; and 3) final approval of the version to be published.

- Acknowledgements

- Funding statement

on behalf of Kevin Padian (Subject Editor)
openscience@royalsociety.org

Associate Editor Comments to Author:

Ordinarily, the journal does not grant authors multiple opportunities to revise their paper(s); however, where 'good faith' attempts at improvement seem to be made by the authors in response to extensive feedback, a degree of flexibility is sometimes permitted. As the Editors consider you to have tried to respond sincerely (if not satisfactorily) to the reviewers, a further opportunity to revise is being made available - you should work hard to tackle constructively the remaining concerns. If you do not do so, and the more critical of the reviewers remains unsatisfied by your efforts, we will not be able to consider further revisions. Good luck.

Subject Editor Comments to Author:

I support the AE's comments and wish you the best in your revisions.

Reviewer comments to Author:

Reviewer: 2

Krashennikova and colleagues have revised their extremely interesting paper on prosocial behaviour in Grey parrots. They examine multiple possible situations and test the birds extensively. I nevertheless still have several comments – some involving minor clarification, others a bit more substantive – which I present in order of appearance.

L. 29: I'd be more comfortable with something like “..their lack of understanding of the contingencies of the particular tasks used in this study, the underlying...” [that is the most that the authors can claim]

L. 39: This paper is on pro-sociality, which the authors just described as being different from altruism. So where does this sentence with respect to altruism fit? Do they mean pro-sociality? Or at least reciprocal altruism?

L. 58-59: The edit is odd...leave out comma after “familiar”...and maybe “and in situations without a possibility...”

L. 118: Remove comma after “While”...Based on the material in the paper, maybe the description should read “did not show clear reciprocity...” Again, read my comments based on the material in the paper cited: When the dominant bird led, he STARTED OUT prosocial but switched to selfish when his partner consistently acted in a selfish manner”...what is written in this manuscript is not correct.

This description of the material is not correct. Citing from the paper itself: “However, whether they shared with their partner at no cost to themselves seemed, at least for our subjects, to depend upon their dominance status and upon who led and who followed. Griffin has always been dominant to Arthur, and Figs. 3 and S1 show that when Griffin led, he became selfish, but when placed in the position of follower, he acted in a more sharing manner.” That is, when Griffin led he started out sharing but when Arthur became more selfish, Griffin became more selfish. When Griffin followed, he retained his more sharing manner. Arthur, the subordinate, “..started with a slight giving tendency as both leader and follower, then became statistically significantly more selfish both as leader and follower.”

L. 120ff: THEREFORE, these lines should read: “If the dominant bird started the session choosing first, he initially chose the prosocial token, but became more selfish after he experienced selfish behavior from his partner. When the dominant bird was the follower, he retained his prosociality. The subordinate bird, however, became more selfish over time in both leader and follower roles. Therefore, both birds developed...”

L. 125: This sentence is not correct – and the sentence has not been altered....I repeat: the authors tested for side biases and the current authors cannot disregard these tests. I will change my decision to “reject” rather than revise if this revision is not made in a subsequent version.

L. 126: This sentence is also not correct. Control was inherent in the task: Note that the birds did not copy the human, as the birds were not giving treats to the generous human but sharing treats, and they quickly learned to avoid the null cup (no treats to anyone), demonstrating that they clearly understood the task. Note, too, the follow-up study that further tested the task's assumptions. The authors of the current paper cannot blithely disregard data and criticize another paper without full justification. Avoiding the null cup if it gives no reward can be described any way you choose, but it still involves understanding the contingencies of the task. And explaining the choice of the option that provides two rewards rather than one as

reinforcement learning is meaningless unless the chooser GETS more rewards. If the chooser gets to share, that means the chooser understands the contingencies of the choice. As for the control without a partner – the subject is simply likely going to be confused about what is going on – see more on that later.

The argument about color bias is justified, however.

L. 134: I think the authors mean “employed” rather than “applied”

L. 201: “without shells”

L. 199: This sentence is not clearly written. Try dividing the sentence into two. “...in all equal reward conditions. When the actor chose prosocially in the unequal condition, in contrast, the partner received a piece of walnut (preference for walnut was assessed prior to the experiment) while the actor received merely the same sunflower seed as in the equal condition.”

L. 217: I think it is important to note here that these birds did NOT express inequity aversion previously. Such finding suggest that the authors already know that the birds are not that likely to be affected by their partner getting a better reward, and that this is a further test of that finding.

L. 216: In the response to reviewers, the authors mention that they did not find any position preference in token choice...this information should be provided somewhere in the results section.

L. 226: Is this a new paragraph?

L. 244: Sentence a bit confusing. Divide sentences. “The rewards inside the cups were not at that time visible to the birds. Once a choice was made, the experimenter lifted the lid(s) off the small food container(s), tilted the cup(s) forward so that both birds could see the reward(s) inside, and then distributed the reward(s).”

L. 259ff: If birds don’t care about inequality, why would they care if one got fed and one didn’t? Isn’t that finding a problem for this entire paper? The receiver could eventually associate the ring color with whether it gets food, but not necessarily with the actions of its partner if it doesn’t care that its partner gets fed and it doesn’t, especially if it was going to get at least half the standard reward just for hanging out near the reward site when the experimenter wanted it to move back. If a whole seed is a standard reward, half that reward is hardly “tiny” [and, yes, I do understand that sunflower seeds cannot be divided into much smaller pieces and still easily be handled, but maybe the authors should have used some other, less favored food?]

L. 270: 12 sessions needed to be repeated? I’m confused...there are only 9 sessions listed in Table 1... If this statement is not a typo, then there are a LOT of repetitions. Could the authors please let the reader know how many sessions were repeated for each condition? That would help the reader know which conditions were more problematic. The authors gave some of the information in their response but not in the paper...it seems as though one dyad really had issues, and it would be helpful to know which one. Maybe that set of data needs to be dealt with separately?

L. 273: What happened after the birds chose a token in the color preference test? Did it just get taken away? Did they get something or nothing for making the choice? If they had any experience in trading tokens for treats, and they chose a token and got nothing, they would – if they had any sense – alternate choice of tokens to see if choosing a different token would give a reward, even if they liked their first choice. It seems that most birds started with a color

preference and those colors were eliminated, but it also seems possible that the birds just LEARNED at this point that they should choose randomly because nothing would happen almost whatever they did BUT if they chose their favorite it would disappear. I don't disagree that one has to check for color preferences, but it's a rather tricky procedure as they learn to ignore any contingencies connected with the color.

L. 286ff: So now a bird was given "x" to choose and got a reward half the time and was given "y" to choose the other half and still got a reward; sometimes their partner got a reward and sometimes the partner did not. Given that they didn't care about reward inequality, why would they pay much attention to what was happening, particularly if they were actor but even if they were the powerless recipient? If the recipient couldn't do anything about what happens, and can't tell the actor what to do, why pay too much attention?

L. 298: And I'm still confused...The authors mention equal and unequal rewards earlier, but Table 1 does not, so I'm confused. Does each bird have one session as actor with equal rewards then one with unequal reward before the roles are reversed (i.e., that's what is meant by 2 sessions in Table 1)? Or do reversals occur after equal reward and then again before unequal reward? And do all these trials occur before the alternating sessions? I need to understand what is happening at this point in the paper. Or do the birds cycle through UNI, ALT, YOK in the equal reward before doing unequal rewards? [It seems that way, given that different tokens were used, but I really had to work to figure this out and I'm still not sure.] If I were trying to replicate this experiment, I'd never be able to figure out exactly what was done. The material now taken out helped me before...

L. 305: Two sessions EACH? Including equal or unequal rewards?

L. 343: Do the authors mean 30 trials per bird acting as actor and 30 as recipient? And where do the equal/unequal fit in here...one session with equal rewards, one session with unequal rewards? Not clear...

L. 355ff: I absolutely do not understand the authors' response to my query. Why would switching to a selfish strategy be the expected response? The experimenter is NOT always being selfish, so why would the actor be selfish? The authors are making AN ASSUMPTION that the birds are treating the experimenter as an automaton, and ignoring the possibility that they were playing with the experimenter. In previous studies, didn't the experimenter played partner-like roles? So why ignore that now? Given that there were lots of prosocial choices in the alternating condition, the experimenter was being prosocial a lot, so why wouldn't the actor reciprocate to that? I definitely appreciate the new material added around L. 580.

L. 384: Accessible vs inaccessible: So, I'm a bird and you train me to go back and forth between compartments. I learn that I can get food in both places, but not with respect to anything to do with token choice; I just learn that it's possible. And there's no bird next door. If I'm Jelo or Bella or Sensi or Nikki (half the birds), I don't notice whether or not the passage is open but I choose to put food on the other side because I remember that it's possible to get food. If I'm Kizzi or Nina or Jack, whether or not the passage is open I'm not prosocial maybe because there's no bird there. If I'm Kimmi, I'm just confused. Actually, if the birds differentiated between the conditions, they should NOT be random but choose selfish in inaccessible because they'd otherwise be wasting food, and they should choose prosocial in the accessible condition to get more food. But none of the birds (except a tendency in Jack, in the opposite to what would be expected) seemed to understand the different conditions at all. They seemed to hit on some strategy simply with respect to the possibility of more food — hope for the best or accept the worst — and stick to it. I don't see the task as showing much of anything about understanding cooperation. I think the authors' comments in the Discussion about what they did and didn't learn and spatial confusion

make more sense than the assumption that the task would be useful, particularly given that the birds' behaviours altered over time.

Re blockage...about half the birds seemed to understand the issue and the other half did not.

Overall: Individual differences seem to be extremely important in this study, from looking at the supplemental data. Maybe some Greys are good buddies and others simply are not, or maybe some birds are dim and others not...the authors really need to look at and discuss the individuals, not just the group behaviour.

L. 451ff: I'm confused...if birds DID develop preferences, and there were so many individual differences in the results, shouldn't the experimenters have checked specifically to see if the birds' actions were affected by these preferences? Just looking at group behaviour is not sufficient, as the individuals varied so strongly.

L. 474: The authors must mean dashed red line, not dotted....I'm not sure what the dotted line means.

L. 560: Correct English is "fewer prosocial", not "less prosocial"...

We still do not know the relationships between the individuals in the pairs, although the authors do admit to the artificiality of the situation. Note: A 'good' relationship doesn't necessarily mean equal; if the birds have worked out a dominant-subordinate relationship and are happy with it, that makes it "good"...but if there is conflict, that is a different situation, and could easily affect the results.

We also know that parrots engage in courtship feeding, where one bird gives up food to the other with no reciprocity, and in Greys both parents feed the young directly from what they obtain when they forage (unlike mammals, who suckle, which is quite different in terms of sharing resources)...so the whole idea of unequal reward as an issue is quite different from what one would see in primates.

I'm still very conflicted about this paper – given how critical the experimenters are of the previous studies, their failure to examine the material of the previous studies carefully, and their claims that they performed their experiments in order to correct all the problems they perceived in the previous studies, they really have to be exceptionally careful about what they are saying and claiming. I re-read the paper even more carefully this time, and found other inconsistencies that need to be addressed. I'm not trying to be obnoxious – I don't want someone else to take them apart after the paper is published. I think that looking at the behaviour both of individuals and individual pairs would be very helpful.

Again, the good part of this paper is that the authors are honest in their overall evaluation that additional work is necessary to determine prosociality in parrots (at least Grey parrots). And, yes they did test more birds than in previous studies, but looking at the individual data it seems that some birds are good at being prosocial, others are not, and some learn more about the situation than do others. And I repeat that their controls were confusing...if I had been in the birds' place, I would have been confused. Just because someone else used these controls doesn't mean they are correct. Thus, again, the authors have to be extremely careful about their analyses and claims as well as criticisms.

Reviewer: 1

Comments to the Author(s)

I sincerely thank the authors for their thoughtful responses to my earlier concerns. At this point I have reviewed the revised manuscripts and have no further questions or concerns to raise.

Author's Response to Decision Letter for (RSOS-190696.R1)

See Appendix B.

RSOS-190696.R2 (Revision)

Review form: Reviewer 2

Is the manuscript scientifically sound in its present form?

Yes

Are the interpretations and conclusions justified by the results?

No

Is the language acceptable?

Yes

Do you have any ethical concerns with this paper?

No

Have you any concerns about statistical analyses in this paper?

No

Recommendation?

Accept with minor revision (please list in comments)

Comments to the Author(s)

Krashennikova and colleagues have re-revised their extremely interesting paper on prosocial behaviour in Grey parrots. They examine multiple possible situations and test the birds extensively. I nevertheless still have several comments – some involving minor clarification, others a bit more substantive – which I present in order of appearance.

L. 75: Authors might be interested in a new paper: PLoS One: Female rats release a trapped cagemate following shaping of the door opening response: Opening latency when the restrainer was baited with food, was empty, or contained a cagemate
Magnus H. Blystad , Danielle Andersen, Espen B. Johansen

Also: Should read “methodologies”

L. 104: Odd grammar: Probably should read something like “Although few studies have focused on...in the wild, a large number of studies...in the laboratory.”

L. 124: The authors again do not attend to the studies. They are confusing ref. 53 and ref. 54. In a second experiment in *Iref. 53*, the human was either generous or selfish and the parrots modulated their responses accordingly. In the follow-up study in *ref. 54*, the human exactly mirrored the parrot’s behavior, and the parrot learned the “tit-for-tat” response quickly. So...if the authors want to ignore the human-parrot study (ref. 54) to some extent because it is not parrot-parrot, that is a separate issue, but they should not confound the two studies....

L. 129: This description of the material is STILL not correct. Again, I am truly annoyed at the authors’ inability to separate the second experiment in ref. 53 from the study in ref. 54. In ref. 54, citing from the paper itself: “Cup positions were now changed after every human choice, with positions determined by use of the program random.org.”

The point is that the authors did check for side biases in their subsequent study...and there were NO effects... If the same bird did not have position preferences in ref. 54, it did not have them in ref. 53.

According to the authors in ref. 54, that subsequent study was also designed to examine the parrot’s understanding of the task...would the parrot learn that the recipient was playing tit-for-tat and respond in kind? So, again the statement with respect to understanding is not valid, and the citation of ref. 24 also is with respect to ref. 53 (ref. 24 does not even cite ref. 54, where the bird was the first actor).

The argument about color bias is justified. The current authors maybe should just stop there, or clarify that their problem with refs. 53 and 54 was that no control existed to determine how the subjects would act in the absence of partners.

L. 139: The third objective is the same as in the second experiment of ref. 53 and was central to ref. 54.

L. 146: Grammar – “We varied the reward distribution, using “equal” or “unequal” rewards, to test....distribution (i.e., making....)” [I believe a parenthesis is missing]

L. 154: “...chose the prosocial...”

L. 198, 199: “When the actor...”

L. 226: “...overlap but were still...”

L. 253: How many is “a few”?

L. 268: The authors’ response to the reviewer comments was a lot clearer than what they wrote in the manuscript.... *In the colour preference test, any choice was rewarded in the same way. Two different tokens were presented in one session in a shallow box. After the bird chose one token, it was rewarded with a sunflower seed and the trial ended. The next trial started after a 10 sec interval. Each subject was offered 12 choices, and if any token was chosen 10 times or more by one of the two birds of one pair, the token would be replaced by another colour in the next session.* This additional information about reward would be extremely helpful, so that readers understand what the bird actually experienced.

L. ~276ff: In the familiarization condition, a bird was given “x” to choose and got a reward half the time and was given “y” to choose the other half and still got a reward; sometimes their

partner got a reward and sometimes the partner did not. I repeat....why would they pay much attention to what was happening, particularly if they were actor but even if they were the powerless recipient? If the recipient couldn't do anything about what happens, and can't tell the actor what to do, why pay too much attention? That's what happened in the UNI condition...The authors' response makes it clear that the birds care in the ALT condition, in which they are NOT powerless:*In reply to the reviewer's last 2 sentences...we have also evidence that even recipients pays attention to the reward contingencies in both compartments as revealed by the increase in prosocial choices in our ALT (alternating) condition. In the ALT condition, the recipients were not "powerless" because they had the opportunity to reciprocate/retaliate in an alternating manner.....which is exactly my point...there is no reason to pay attention unless they can take an active role.*

The authors have to clarify WHY the UNI condition is useful earlier on (they do so in L. 307, but until then a reader likely thinks it's a totally useless task)... In the Discussion, they should emphasize the point that in the UNI condition, birds' failures to respond as expected could have been because they did not receive feedback from the recipient, who was powerless to respond, and because the recipient might not even have attended to the behaviour of the actor in the UNI condition for that reason.

L. 327ff: The authors seem not to understand my criticism. My point: The authors are making AN ASSUMPTION that the birds are treating the experimenter as an automaton, and ignoring the possibility (at least until the Discussion) that they were playing the game with the experimenter. The issue is not only whether the actor understood that the recipient could not choose, but also UNDERSTOOD that the experimenter was NOT deliberately choosing...How could the authors be sure of the parrot's interpretation? They suggest this possibility in the Discussion, but it needs to be made clearer.

L. ~360ff: Again, the authors seemed not to understand my criticism....I'm basically questioning the usefulness of Accessible vs Inaccessible: So, I'm a bird and you train me to go back and forth between compartments. I learn that I can get food in both places, but NOT with respect to anything to do with TOKEN choice; I just learn that it's possible when, for some odd reason, there's no bird next door. If I'm Jelo or Bella or Sensi or Nikki (half the birds), I don't notice whether or not the passage is open but I choose to put food on the other side because I remember that it's possible to get food. If I'm Kizzi or Nina or Jack, whether or not the passage is open I'm not prosocial maybe because there's no bird there. If I'm Kimmi, I'm just confused. None of the birds (except a tendency in Jack, in the opposite to what would be expected) seemed to understand the different conditions at all. They seemed to hit on some strategy simply with respect to the possibility of more food – hope for the best or accept the worst – and stick to it. I don't see the task as showing much of anything about understanding cooperation. Or...Maybe the birds kept hoping their actions would eventually help their partner, whom they expected to return because the partner had been there for all the other types of trials?

What about a different control scenario that would not be confusing to the bird?...What if the recipient had plenty of food and didn't NEED anything from the actor, would the actor have continued to be prosocial?

L. 395: Again, I see this task as simply confusing to the birds, not an actual test of their understanding...

L. 511: "...although such tendencies..."

L. 515ff, 586ff, 642: The issue to me is that the task contingencies are inherently confusing...which is the point of my criticisms above. I understand that the authors wanted to replicate work with nonhuman primates, but I simply don't think that some of the tasks actually tested what they were purportedly designed to test. Parrots may be inherently less competitive than most

nonhuman primates; tasks that might make sense to nonhuman primates might not make sense to parrots...Again, I understand that the authors wanted direct comparisons with nonhuman primates, but then they need to spend more space in the Discussion (not just a brief mention) explaining why their data differed, based on the ecology and ethology of parrots compared to nonhuman primates.

And, as some dyads were just more prosocial than others...one of the in-depth issues for the Discussion is the possibility that individual relationships may be more important to parrots than nonhuman primates and that trying to examine the issue on a group level may be more difficult than it might be in nonhuman primates...That is, maybe if these individual relationships are so important, the birds will *respond* in ways that differ from how nonhuman primates respond?

L. 550: Yes...

L. 557: Maybe the UNI prosocial walnut choice was some attempt to get the receiver to attend to what was happening? Given that the UNI unequal followed the ALT equal, maybe they thought that somehow if the recipient got a special treat, s/he would respond, breaking the UNI condition?

L. 668: I don't think the issue was the lack of cognitive resources in the parrots as much as the confusing conditions of the task.

Again, the good part of this paper is that the authors are honest in their overall evaluation that additional work is necessary to determine prosociality in parrots (at least Grey parrots). And, yes they did test more birds than in previous studies, but looking at the individual data it seems that some birds are good at being prosocial, others are not, and some learn more about the situation than do others. And I repeat that their controls were confusing...if I had been in the birds' place, I would have been confused.

What I'm trying to get at here is that the authors made an excellent, good-faith attempt at using a protocol that was designed for nonhuman primates, with very limited revision, for Grey parrots. They found that when the birds could *actually* interact with one another, some pairs acted prosocially, other pairs (like the pair in ref. 53), did not. They also found that the controls may not have been appropriate for the parrots...yes, the tasks might have been beyond the parrots' processing abilities, but given how many other tasks these birds solve appropriately, it is more likely that the birds "overthought" the tasks and responded in ways more appropriate to their ecology/ethology.

In sum....A lot of what the authors wrote in response to the reviewer comments would be a terrific part of the Discussion, and I hope that the editors allow the authors another round so that they can insert that material. My only disagreement with the following is "unintended variables" – the issue is more likely different types and degrees of species-specific social interactions in parrots and nonhuman primates...the authors almost get to it in the current version, but what they write in their rebuttal would be a lovely end to their paper: *it is the overarching dilemma of animal cognition studies, that we do not know how the test animals under investigation perceive a test situation. Most studies aim at designing their studies as ethologically valid and salient for the species under investigation as possible, but often it turns out retrospectively that unintended variables affected the animals in the test situation. This is a common pitfall of comparative cognitive studies in particular where one strives for maximum comparability by keeping the experimental protocols as similar as possible.*

Decision letter (RSOS-190696.R2)

04-Nov-2019

Dear Dr Krasheninnikova:

On behalf of the Editors, I am pleased to inform you that your Manuscript RSOS-190696.R2 entitled "Assessing African grey parrots' prosocial tendencies in a token choice paradigm" has been accepted for publication in Royal Society Open Science subject to minor revision in accordance with the referee suggestions. Please find the referees' comments at the end of this email.

The reviewers and Subject Editor have recommended publication, but also suggest some minor revisions to your manuscript. Therefore, I invite you to respond to the comments and revise your manuscript.

- Ethics statement

- Data accessibility

If you wish to submit your supporting data or code to Dryad (<http://datadryad.org/>), or modify your current submission to dryad, please use the following link:
<http://datadryad.org/submit?journalID=RSOS&manu=RSOS-190696.R2>

- Competing interests

- Authors' contributions

- Acknowledgements

- Funding statement

Because the schedule for publication is very tight, it is a condition of publication that you submit the revised version of your manuscript before 13-Nov-2019. Please note that the revision deadline will expire at 00.00am on this date. If you do not think you will be able to meet this date please let me know immediately.

Supplementary files will be published alongside the paper on the journal website and posted on

the online figshare repository (<https://figshare.com>). The heading and legend provided for each supplementary file during the submission process will be used to create the figshare page, so please ensure these are accurate and informative so that your files can be found in searches. Files on figshare will be made available approximately one week before the accompanying article so that the supplementary material can be attributed a unique DOI.

Kind regards,
 Andrew Dunn
 Senior Publishing Editor
 Royal Society Open Science
 openscience@royalsociety.org

on behalf of Prof Kevin Padian (Subject Editor)
 openscience@royalsociety.org

Associate Editor Comments to Author:

The reviewer has presented a number of critiques of this paper through its iterations, and the Editors would like to thank them for the guidance and also encourage the authors to constructively engage with these critiques. Please do ensure that you take the time to make the changes requested, and present these in a tracked changes version of the paper, along with a full point-by-point response. This will help the Editors determine whether the reviewer needs to be consulted further.

Reviewer comments to Author:

Reviewer: 2

Comments to the Author(s)

Krasheninnikova and colleagues have re-revised their extremely interesting paper on prosocial behaviour in Grey parrots. They examine multiple possible situations and test the birds extensively. I nevertheless still have several comments – some involving minor clarification, others a bit more substantive – which I present in order of appearance.

L. 75: Authors might be interested in a new paper: PLoS One: Female rats release a trapped cagemate following shaping of the door opening response: Opening latency when the restrainer was baited with food, was empty, or contained a cagemate
 Magnus H. Blystad , Danielle Andersen, Espen B. Johansen

Also: Should read “methodologies”

L. 104: Odd grammar: Probably should read something like “Although few studies have focused on...in the wild, a large number of studies...in the laboratory.”

L. 124: The authors again do not attend to the studies. They are confusing ref. 53 and ref. 54. In a second experiment in *ref. 53*, the human was either generous or selfish and the parrots modulated their responses accordingly. In the follow-up study in *ref. 54*, the human exactly mirrored the parrot’s behavior, and the parrot learned the “tit-for-tat” response quickly. So...if

the authors want to ignore the human-parrot study (ref. 54) to some extent because it is not parrot-parrot, that is a separate issue, but they should not confound the two studies....

L. 129: This description of the material is STILL not correct. Again, I am truly annoyed at the authors' inability to separate the second experiment in ref. 53 from the study in ref. 54. In ref. 54, citing from the paper itself: "Cup positions were now changed after every human choice, with positions determined by use of the program random.org."

The point is that the authors did check for side biases in their subsequent study...and there were NO effects... If the same bird did not have position preferences in ref. 54, it did not have them in ref. 53.

According to the authors in ref. 54, that subsequent study was also designed to examine the parrot's understanding of the task...would the parrot learn that the recipient was playing tit-for-tat and respond in kind? So, again the statement with respect to understanding is not valid, and the citation of ref. 24 also is with respect to ref. 53 (ref. 24 does not even cite ref. 54, where the bird was the first actor).

The argument about color bias is justified. The current authors maybe should just stop there, or clarify that their problem with refs. 53 and 54 was that no control existed to determine how the subjects would act in the absence of partners.

L. 139: The third objective is the same as in the second experiment of ref. 53 and was central to ref. 54.

L. 146: Grammar – "We varied the reward distribution, using "equal" or "unequal" rewards, to test....distribution (i.e., making....)" [I believe a parenthesis is missing]

L. 154: "...chose the prosocial..."

L. 198, 199: "When the actor..."

L. 226: "...overlap but were still..."

L. 253: How many is "a few"?

L. 268: The authors' response to the reviewer comments was a lot clearer than what they wrote in the manuscript.... *In the colour preference test, any choice was rewarded in the same way. Two different tokens were presented in one session in a shallow box. After the bird chose one token, it was rewarded with a sunflower seed and the trial ended. The next trial started after a 10 sec interval. Each subject was offered 12 choices, and if any token was chosen 10 times or more by one of the two birds of one pair, the token would be replaced by another colour in the next session.* This additional information about reward would be extremely helpful, so that readers understand what the bird actually experienced.

L. ~276ff: In the familiarization condition, a bird was given "x" to choose and got a reward half the time and was given "y" to choose the other half and still got a reward; sometimes their partner got a reward and sometimes the partner did not. I repeat....why would they pay much attention to what was happening, particularly if they were actor but even if they were the powerless recipient? If the recipient couldn't do anything about what happens, and can't tell the actor what to do, why pay too much attention? That's what happened in the UNI condition...The authors' response makes it clear that the birds care in the ALT condition, in which they are NOT powerless:*In reply to the reviewer's last 2 sentences...we have also evidence that even recipients pays attention to the reward contingencies in both compartments as revealed by the increase in prosocial choices*

in our ALT (alternating) condition. In the ALT condition, the recipients were not “powerless” because they had the opportunity to reciprocate/retaliate in an alternating manner.....which is exactly my point...there is no reason to pay attention unless they can take an active role.

The authors have to clarify WHY the UNI condition is useful earlier on (they do so in L. 307, but until then a reader likely thinks it's a totally useless task)... In the Discussion, they should emphasize the point that in the UNI condition, birds' failures to respond as expected could have been because they did not receive feedback from the recipient, who was powerless to respond, and because the recipient might not even have attended to the behaviour of the actor in the UNI condition for that reason.

L. 327ff: The authors seem not to understand my criticism. My point: The authors are making AN ASSUMPTION that the birds are treating the experimenter as an automaton, and ignoring the possibility (at least until the Discussion) that they were playing the game with the experimenter. The issue is not only whether the actor understood that the recipient could not choose, but also UNDERSTOOD that the experimenter was NOT deliberately choosing...How could the authors be sure of the parrot's interpretation? They suggest this possibility in the Discussion, but it needs to be made clearer.

L. ~360ff: Again, the authors seemed not to understand my criticism....I'm basically questioning the usefulness of Accessible vs Inaccessible: So, I'm a bird and you train me to go back and forth between compartments. I learn that I can get food in both places, but NOT with respect to anything to do with TOKEN choice; I just learn that it's possible when, for some odd reason, there's no bird next door. If I'm Jelo or Bella or Sensi or Nikki (half the birds), I don't notice whether or not the passage is open but I choose to put food on the other side because I remember that it's possible to get food. If I'm Kizzi or Nina or Jack, whether or not the passage is open I'm not prosocial maybe because there's no bird there. If I'm Kimmi, I'm just confused. None of the birds (except a tendency in Jack, in the opposite to what would be expected) seemed to understand the different conditions at all. They seemed to hit on some strategy simply with respect to the possibility of more food – hope for the best or accept the worst – and stick to it. I don't see the task as showing much of anything about understanding cooperation. Or...Maybe the birds kept hoping their actions would eventually help their partner, whom they expected to return because the partner had been there for all the other types of trials?

What about a different control scenario that would not be confusing to the bird?...What if the recipient had plenty of food and didn't NEED anything from the actor, would the actor have continued to be prosocial?

L. 395: Again, I see this task as simply confusing to the birds, not an actual test of their understanding...

L. 511: "...although such tendencies..."

L. 515ff, 586ff, 642: The issue to me is that the task contingencies are inherently confusing...which is the point of my criticisms above. I understand that the authors wanted to replicate work with nonhuman primates, but I simply don't think that some of the tasks actually tested what they were purportedly designed to test. Parrots may be inherently less competitive than most nonhuman primates; tasks that might make sense to nonhuman primates might not make sense to parrots...Again, I understand that the authors wanted direct comparisons with nonhuman primates, but then they need to spend more space in the Discussion (not just a brief mention) explaining why their data differed, based on the ecology and ethology of parrots compared to nonhuman primates.

And, as some dyads were just more prosocial than others...one of the in-depth issues for the

Discussion is the possibility that individual relationships may be more important to parrots than nonhuman primates and that trying to examine the issue on a group level may be more difficult than it might be in nonhuman primates...That is, maybe if these individual relationships are so important, the birds will *respond* in ways that differ from how nonhuman primates respond?

L. 550: Yes...

L. 557: Maybe the UNI prosocial walnut choice was some attempt to get the receiver to attend to what was happening? Given that the UNI unequal followed the ALT equal, maybe they thought that somehow if the recipient got a special treat, s/he would respond, breaking the UNI condition?

L. 668: I don't think the issue was the lack of cognitive resources in the parrots as much as the confusing conditions of the task.

Again, the good part of this paper is that the authors are honest in their overall evaluation that additional work is necessary to determine prosociality in parrots (at least Grey parrots). And, yes they did test more birds than in previous studies, but looking at the individual data it seems that some birds are good at being prosocial, others are not, and some learn more about the situation than do others. And I repeat that their controls were confusing...if I had been in the birds' place, I would have been confused.

What I'm trying to get at here is that the authors made an excellent, good-faith attempt at using a protocol that was designed for nonhuman primates, with very limited revision, for Grey parrots. They found that when the birds could *actually* interact with one another, some pairs acted prosocially, other pairs (like the pair in ref. 53), did not. They also found that the controls may not have been appropriate for the parrots...yes, the tasks might have been beyond the parrots' processing abilities, but given how many other tasks these birds solve appropriately, it is more likely that the birds "overthought" the tasks and responded in ways more appropriate to their ecology/ethology.

In sum....A lot of what the authors wrote in response to the reviewer comments would be a terrific part of the Discussion, and I hope that the editors allow the authors another round so that they can insert that material. My only disagreement with the following is "unintended variables" – the issue is more likely different types and degrees of species-specific social interactions in parrots and nonhuman primates...the authors almost get to it in the current version, but what they write in their rebuttal would be a lovely end to their paper: *it is the overarching dilemma of animal cognition studies, that we do not know how the test animals under investigation perceive a test situation. Most studies aim at designing their studies as ethologically valid and salient for the species under investigation as possible, but often it turns out retrospectively that unintended variables affected the animals in the test situation. This is a common pitfall of comparative cognitive studies in particular where one strives for maximum comparability by keeping the experimental protocols as similar as possible.*

Author's Response to Decision Letter for (RSOS-190696.R2)

See Appendix C.

Decision letter (RSOS-190696.R3)

17-Nov-2019

Dear Dr Krashennikova,

It is a pleasure to accept your manuscript entitled "Assessing African grey parrots' prosocial tendencies in a token choice paradigm" in its current form for publication in Royal Society Open Science. The comments of the reviewer(s) who reviewed your manuscript are included at the foot of this letter.

on behalf of Prof Kevin Padian (Subject Editor)
openscience@royalsociety.org

Appendix A

Response to reviewers

Reviewer: 1

Comments to the Author(s)

The manuscript entitled “Assessing African grey parrots’ prosocial choice tendencies in a token choice paradigm” presents findings from a prosocial choice task in African grey parrots. The study is a thorough replication of previous work done in primates and as such, provides a nice point for comparison in the literature. Overall the methodology was well thought out, the analyses appropriate, the literature review thorough, and the conclusions were sound. I have one major concern with the methods, but all other suggestions or questions are minor, denoted by line numbers below.

We thank the reviewer for this feedback.

Introduction

Line 58: This seems like a bit of an exaggeration, given that there are ample studies that show cooperative behavior among non-kin in primates in the absence of reciprocation. For example, in studies [20,25] there was no opportunity for reciprocity within the experiment and the primates still were cooperative. I suspect the authors are suggesting that nonkin cooperate only when they are familiar (which has been difficult to study since it is challenging to find species and settings where strangers can be tested together). But this line seems like a bit of an overstatement as it is written, especially since in [20] both kin and non-kin pairs behaved similarly.

Response: *We agree and have changed the wording accordingly.*

Line 57ff: *“Indeed, cooperative behaviours in non-human primates seem to be directed more frequently to kin and to familiar, hence reciprocating, partners [17, 18], but have also been reported between non-kin partners and without a possibility for reciprocity [20, 25].”*

Line 108: While the parrots succeeded at the string-pulling task in [49], they did not show awareness that a partner was needed in the delay condition. By most accounts, this would be considered a failure at cooperation, since they didn’t appear to notice the role of the partner.

Response: *In the Péron et al. (2011) study, the parrots were able to coordinate their actions in order to pull a platform with food into reach. You are correct in stating that they failed in the delayed control (in which they needed to inhibit pulling the string until the partner arrived); however, when given the choice between two apparatuses – one which could be solved alone and one which required a partner - they correctly chose the appropriate apparatus. Consequently, indicating that the parrots were able to take the necessity for a partner into account, but potentially due to a lack of inhibitory control, they could not refrain from pulling the string when no partner was available (as in the delayed control). Nonetheless, we have changed the wording in the manuscript and no longer talk about cooperative abilities but rather the ability to coordinate actions (line 109).*

Methods

General comment: I am sure this was unintentional, but at times the language follows Suchak & de Waal [ref 21] extremely closely, even verbatim. I recognize that when doing a replication it may be hard to think of alternative ways to phrase things, but the authors need to extensively re-word large portions of the methods section to avoid violating copyright. For this reason I am suggesting a major

revision.

Response: Thank you for noticing this overlap in the descriptions between studies. We did so unintentionally and have changed the wording of the methods.

Line 213: Why were the compartment sides fixed, rather than counterbalanced across days/sessions? Could this possibly have inhibited them from moving from side to side in the accessible condition?

Response: The reason for fixed compartments sides was to ensure that the subjects “knew” their role across the conditions, i.e., who is the choosing individual. Since we provide them with a training session before starting the accessible condition during which they had to move from side to side, it is unlikely that the fixed compartments might have inhibited them from moving from walking towards the other side: only once after they had changed the sides 12 times in a row they proceeded with the test condition. We added this information in the revised version of the manuscript (line 224ff).

Line 220: I think this should say the partner could witness the actor’s behavior

Response: Thank you for noticing this mistake, we have changed it accordingly.

Results:

Line 421: One possibility for this, which hasn’t really been addressed in the discussion is that during the unequal ALT condition, the birds might experience a frustration effect, since in one trial they might get a high value reward as the partner, but in the next get a low value reward as the actor, even though the token chosen in both trials is the same.

Response: This is an interesting interpretation that we have not thought of so far. We have added this to the discussion.

Line 590ff: “The decrease of prosocial choices in the unequal ALT compared to the equal ALT condition could be explained by a frustration effect. The switch from receiving seeds (when in actor role) after having received walnut in the previous trial (if partner made a prosocial choice) might be perceived as frustrating as the reward quality decreased. Potentially, the induced frustration affected the parrots’ other-regarding preferences.”

Line 443: Are the descriptives the $M \pm SD$? If so, can you specify that?

Response: Yes, these results depict the mean \pm SD. We have added this information to the manuscript (l.528).

Line 450: Have you tried a logistic regression using previous choice as a predictor, as opposed to the contingency table? Given that there were differences among individuals/pairs (as discussed in lines 613-616), it seems possible that certain individuals might be driving this effect, which would be masked in pooled data.

Response: We were not able to run a model on the data, as the model assumptions could not be fulfilled (i.e. uncorrectable overdispersion). Consequently, we looked at the contingency on a dyadic level rather than on the population level and found that none of the dyads exhibit significant contingency between choices (see Table S4 for details).

Line 457: I think the interpretation “indicating that the presence of a partner...” does not match the

statistic presented here, since in both UNI and SFC the partner was present. It seems to me that this suggests that they weren't making their choice based on whether or not it was actually helping the partner.

Response: *You are right, the indifference between UNI and SFC actually indicates that prosocial choices might have been caused by social facilitation instead of an understanding of whether help can be provided or not. We changed the sentence accordingly in the manuscript.*

Line 501ff: "The birds' prosocial choices did not differ between the UNI condition and the SFC ($49.5 \pm 39.2\%$; GLMM: $\beta = 0.01$, $SE = 0.15$, $z = 0.08$, $p = 0.940$); consequently, we cannot rule out that the presence of the partner facilitated prosocial choices irrespective of whether they actually helped the partner or not."

Line 464: It seems odd to me that the INACC condition is the only one in which the birds changed over the 10-trial blocks. I note that half of the birds started with ACC and the other started with INACC. Is it possible this effect (and the overall high number of prosocial choices) is driven by the birds that started with ACC, since in the previous condition choosing the prosocial token was advantageous? Perhaps they needed to learn that they could not access both rewards?

Response: *We controlled for an order effect (i.e. whether birds starting with the INACC choose fewer prosocial tokens than birds starting with the ACC) and found no difference between birds. We have added the statistics to the manuscript (line 505ff).*

Discussion:

Line 546: I think there is a sentence structure issue in the sentence starting "While their behaviour..."

Response: *We have clarified the sentence: Line 600ff: "The increase in prosocial choices across sessions in the unequal alternating condition could additionally be explained by reinforcement learning (i.e. pick same token as partner) or attitudinal reciprocity."*

Line 577: I am not necessarily recommending the authors do this (as I understand how difficult it is to run more conditions), but retesting the birds in the partner absent conditions with fresh tokens could identify whether it was the tokens or the set up causing this issue. If they still stayed to one side with fresh tokens, you would know it is the latter explanation, but if they showed more flexibility with fresh tokens, it would likely indicate that they formed fixed preferences for the tokens used in previous conditions.

Response: *We certainly agree with your suggestion and ideally, we would re-test the birds with a new set of tokens, however, considering that 1) the actual test was conducted more than one year ago and 2) a different study using other tokens was conducted in the meantime (thus potentially providing them with additional experiences), we unfortunately believe that re-testing the birds would not give us unbiased information about effects of token preferences for this study.*

General comment: it seems to me that the two PCT studies [the current study and ref. 53] combined with the string pulling task [ref 49], all suggest that the parrots may not be monitoring and responding to their partner's behavior in the same way that we see in other species. This is not to suggest that they cannot truly cooperate by taking into account the partner's behavior [c.f. Ref 64], but perhaps tasks originally designed for primates are not tapping into these abilities in a parrot-friendly way. This is not meant as a criticism of this particular study (how else would we know except

to try tests that have already been designed?) but rather something to consider adding to the discussion as it provides a suggestion for people in the field to think outside the primate box, so to speak.

Response: *Thank you for this remark, we definitely agree. We have added this aspect to the discussion in the manuscript.*

Line 673ff: “While established paradigms (such as the PCT, which was originally developed for testing primates) certainly pose great potential for comparative research, future studies on avian cognition should also employ paradigms that are specifically designed for testing parrots, considering their specific behaviours and limitations and maximizing their ecological validity. This is important in order to investigate the possibility that their performance may be affected by the choice of paradigm and experimental context.”

Reviewer: 2

Comments to the Author(s)

Krasheninnikova and colleagues have written an extremely interesting paper on prosocial behaviour in Grey parrots. They examine multiple possible situations and test the birds extensively. I nevertheless have several comments – some involving minor clarification, others a bit more substantive—which I present in order of appearance.

L. 67: Would be clearer as: “...the various hypotheses that have been proposed to explain...”

Response: *Thank you for the suggestion, we have changed the sentence accordingly.*

L. 69: Should read: “our understanding not only of...” [in contrast to ‘but also’]

Response: *Thank you for the suggestion, we have changed the sentence accordingly.*

L. 84: Would be clearer as “Similar to the findings on primates, those on corvid prosociality are mixed....pattern as yet.”

Response: *Thank you for the suggestion, we have changed the sentence accordingly.*

L. 86: For the same reason: “...other paradigms, jackdaws (*Corvus monedula*), for example, show...”

Response: *Thank you for the suggestion, we have changed the sentence accordingly.*

L. 94: Should read: “...also making them interesting...candidates in which to investigate...”

Response: *Thank you for the suggestion, we have changed the sentence accordingly.*

L. 100: “...they likely form...”

Response: *Thank you for the suggestion, we have changed the sentence accordingly.*

L. 114: to be accurate, the line should read “...chose among four options that could reward...”

Response: *Thank you for the suggestion, we have changed the sentence accordingly.*

L. 117ff: This description of the material is not correct. Citing from the paper itself: “However,

whether they shared with their partner at no cost to themselves seemed, at least for our subjects, to depend upon their dominance status and upon who led and who followed. Griffin has always been dominant to Arthur, and Figs. 3 and S1 show that when Griffin led, he became selfish, but when placed in the position of follower, he acted in a more sharing manner.” That is, when Griffin led he started out sharing but as Arthur became more selfish over time, Griffin also became more selfish. When Griffin was the follower, he retained his more sharing manner. Arthur, the subordinate, “..started with a slight giving tendency as both leader and follower, then became statistically significantly more selfish both as leader and follower.”

Response: Thank you for pointing this out. We have incorporated this into the manuscript.

Line 118ff: “While, the parrots did not show reciprocity, they changed their behaviour based on who started as the actor in each session. If the dominant bird started the session choosing first, he selected the selfish option more often, while the behaviour was reversed when roles were changed. Nonetheless, both birds developed a selfish tendency over time.”.

L. 122: Should read “...by decreasing or increasing prosocial choices appropriately”. [Must maintain parallel construction—selfish/decrease; generous/increase]

Response: Thank you for the suggestion, we have changed the sentence accordingly.

L. 123: This sentence is not correct—the authors tested for side biases.

Response: We have toned the sentence down saying that the authors did not control **sufficiently** for side biases in our view. Although the authors state that they controlled for the spatial arrangement of cups on the tray, we are not convinced that this was sufficient to rule out side biases. For example, the spatial arrangement of cups was not consistently and randomly changed (i.e. changed only twice across 24/26 sessions). Furthermore, the authors provided statistics that assessed whether the parrots’ choice differed from chance but not whether the birds actually preferred one cup over another. In order to truly rule out side preferences, as well as colour preferences, it would have been necessary to conduct preference tests before assigning cups to specific locations or reward types.

L. 124: This sentence is also not correct. Control was inherent in the task: Note that the birds did not copy the human, as the birds were not *giving* treats to the generous human but *sharing* treats, and they quickly learned to avoid the null cup (no treats to anyone), demonstrating that they clearly understood the task. Note, too, the follow-up study that further tested the task’s assumptions....

Response: We thank the reviewer for helping us to get this right, however we disagree in this point. In our view, this behaviour could easily be explained by reinforcement learning instead of an understanding of the task. The parrots quickly learn to avoid something that is not rewarding for themselves (null cup) and they preferred to choose an option that provides more rewards (i.e., two pieces under sharing cup). In order to demonstrate an understanding of the task, the bird(s) would have needed to be tested in a non-social condition, in which no partner would be present to accept food rewards. If they would have switched their preference from sharing to selfish, this would serve as an indication for the birds’ understanding of the task.

L. 126: Should read: “objective of the present study was to more fully investigate...”

Response: Thank you for the suggestion. We re-phrased this sentence (l.128)

L. 131: remove “meant to”

Response: Thank you for the suggestion, we have changed the sentence accordingly (l.137).

L. 136: Should read “...that would provide...”

Response: Thank you for the suggestion, we have changed the sentence accordingly (l.142).

L. 140: Would be clearer if read: “..(2) this tendency may be...”

Response: Thank you for the suggestion, we have changed the sentence accordingly (l.146).

L. 173: exchange tokens?

Response: Thank you for the suggestion, we have changed the sentence accordingly (l.179).

L. 206: Could the inequity aversion study have affected these results?

Response: We believe that the birds’ previous experience from the inequity aversion study did not affect the behaviour in this task: 1) We used different tokens (i.e. in texture and colour) and the presentation of tokens was very different (i.e. handing one token to the bird vs. choice between multiple tokens presented in a tray), 2) the inequity aversion study was conducted more than 1 year in advance and the birds participated in many other studies in the meantime, 3) the parrots did not express inequity aversion in the study, thus also did not show overt frustration behaviours.

L. 209: I’m confused....There are three clutches, and all birds are the same age...How many clutches/yr by the same parents? Lizzy and Nina were same-sexed siblings from the same clutch and Nikki and Jack were opposite-sexed siblings from the same clutch; Then Kimmi and Jelo were siblings but from different clutches? Ditto for Bella and Sensei? If the birds are from different clutches by the same parents, one is likely to be older and more dominant...Did the authors test for any type of dominance? In the Péron et al. study, dominance turned out to be important....

Response: The birds are from three different clutches and different parents but all born in the same year (2014). Consequently, they differ only marginally in age by several weeks and months, thus make age effects rather limited. Unfortunately, we did not collect behavioural data on dominance for the current study. Nonetheless, we discuss the potential effect of dominance on prosocial behaviour as a likely explanation for the individual variation in our data (line 671ff.).

L. 214: I’m confused: Counterbalanced with respect to what? If the sides are fixed, what is being counterbalanced? With respect to different experiments?

Response: You are right, we made a mistake. We wanted to express that half of the birds were assigned to one side and the other half to the other side, which of course is implied of birds are tested in dyads. We have deleted this sentence (l.226).

L. 216: Even if tokens were jumbled, did birds have a position preference, separate from color?

Response: No, we did not find any evidence for a position preference.

L. 220: I’m confused...do the authors mean “recipient” in at least one of these places where “actor”

is used?

Response: Thank you for noticing this mistake, we have changed it in the manuscript (l.232).

L. 236: At one level I understand why the actor was fed first, but on another level, isn't that a form of inequality? Wouldn't it be better to be simultaneous? Or did the authors want the actor to see the recipient's behaviour and vice versa?

Response: Yes, as you rightly guessed we wanted to ensure that both birds would attend to their partners' outcome instead of being distracted with their own food reward (if we had fed them simultaneously).

Considering that the recipient was fed within 1-2s, we do not believe that this short delay was causing any perception of inequity in the birds. Furthermore, as we could show in another study, currently under review, African grey parrots do not react to inequity in a token exchange paradigm.

L. 240: I certainly understand why a trial would be cancelled in this instance, but how many times did it happen in each condition? Isn't it an indication of frustration? It would seem to be akin to the choice of the purple cup in Péron et al....

Response: On average no more than 4 failed trials occurred per session. All those trials could be repeated at the end of the session given birds' overall high motivation to participate. We believe that the playful behavior of the subjects enjoying manipulating the tokens rather than frustration was the driving factor for failing some trials.

L. 245: What is a 'tiny' reward, given that one sunflower seed isn't exactly a large reward...? Thirty seeds at any one time is quite a bonanza, however....Did birds' motivation wane toward the end of sessions? Also, L. 247: Again, totally legitimate, but how many times did it happen?

Response: Half of small sunflower seeds (total length of around 0.7 cm) were used as rewards for luring the birds back into position. We did not analyse any variable related to motivation (e.g. latency to make a choice or exchange token), consequently, we can only report our impressions. We felt like the birds' motivation was not decreasing throughout the course of a session, as they consistently came forward at the beginning of each trial and chose tokens very quickly. Twelve sessions needed to be repeated due to a lack of motivation (i.e. either not choosing tokens or not exchanging them). We have added this information to the manuscript (line 268ff).

L. 265: I'm confused...if one session had forced choice trials, what was the other session?

Response: Thank you for noticing that. The birds had two token familiarization sessions and both sessions consisted out of 30 forced trials. We corrected this in the manuscript (l.290f).

L. 281: So, the actor first gets 30 (or 60?) trials in which it gets a reward no matter what it is forced to do, but the experimenter expects that the actor is actually paying attention to what happens to its partner, so that it presumably learns what to do to give the partner a reward...Might only the partner be attending to what is happening, because the choice affects only the partner? So the first bird to be unilateral doesn't care, but the second bird to be unilateral does care, and does pay attention...What I'm asking is whether there's a difference in prosocial behavior depending on order. If the birds paid a lot of attention to what was happening during the familiarization trials to the partner, it shouldn't matter, but did they really make the association?

Response: This is definitely a possible explanation, however, when analyzing only the unilateral condition including the different roles at the beginning (i.e. actor or partner first), we do not find a difference in prosocial behaviour (Wald Chisq Test: $\chi^2 = 0.80$, $df = 1$, $p = 0.370$). Consequently, this indicates that either the birds were paying attention to the reward distribution from beginning on, independent of the roles (actor/partner), or they might have learned the task contingencies (i.e. payoff to partner) only when they could alternately make choices as in the alternating condition.

L. 300: Do the authors mean “consecutive” rather than “subsequent”...i.e., one right after another?

Response: Yes, we meant to say in consecutive sessions. We have changed this mistake in the manuscript (l.329ff).

L. 301: Actually, the actor seems to have 60 free-choice trials, split into two sessions...yes? [Not clear as written...]

Response: Yes, you are right. The birds experienced 30 trials per session and in total 60 trials. We have clarified this in the manuscript.

Line 329ff: “One bird (actor) could choose tokens throughout 30 trials, while the partner could only witness the choices (and receive rewards in case of a prosocial choice) but not interfere (see Video). Two sessions with 30 trials each (in total 60 trials) were conducted before switching the roles”

L. 337: I'm confused....why wouldn't the actor then treat the experimenter as the partner, responding to whatever the partner did? Wouldn't you get roughly the same results? The task is testing whether the bird responds to another bird versus the experimenter, not another bird versus the reward. I understand that the experimenter actually provides the rewards in both cases, but if bird A doesn't see bird B making a choice and only sees the experimenter's actions, then there is an obvious difference. Or maybe bird A thinks that its choice makes the experimenter act twice in some way? [That is...the experimenter distributes treats once based on bird A's choice and then once again without anything else happening...] I would think that this is an extremely confusing condition. Maybe the task would work if it was automated in some way, so that there was no visible actor, but in this case...it is confusing.

Response: We agree, the yoked control condition was potentially confusing for the birds and they might have perceived the reward distribution as rather arbitrarily. Nonetheless, we wanted to incorporate the same conditions as in the original study by Suchak & de Waal (2012) in order to be able to compare the parrots' performance with that of primates. The prediction for this control condition, however, assumes that the birds do not see the experimenter as a partner but rather only as a distributing entity, as they cannot change the choices (as they are being copied from the previous alternating condition) but only accept the rewards. Consequently, switching to a selfish strategy would be an appropriate response to the fixed choices of the experimenter (if indeed the partner's behaviour is facilitating prosocial choices). On the contrary, if the birds are solely attending to the outcome independent of the partner's behaviour, we would expect the birds to not change their prosocial choices and this is exactly what we have observed. Automating the reward distribution would certainly have been an interesting option for removing a potential confounding aspect. Nonetheless, it would first need to be established whether a triadic task – like the token exchange task - would be perceived differently if the birds are only interacting with a non-social computer instead of a human.

L. 362ff: Again, how many times did the birds refuse to enter the empty compartment? Makes a difference in their understanding if they chose prosocially but didn't access the additional treat... And it might make a difference as to which birds started with the inaccessible versus which started with accessible conditions... If the birds started with inaccessible, and saw the experimenter remove the food after 2-3 sec, maybe they assumed it was forbidden and figured that they should not try to access it?

Response: *Indeed, this is an important aspect; however, only one bird (Jelo) did not access the empty compartment during the accessible control (in 84% of prosocial choices she did not consume the second reward). The other birds did always consume both rewards. We are aware of the possibility that this might have affected her performance in the task, nonetheless, Jelo did not have a token preference following the test and we did not want to give her additional training sessions, in order to keep the experience comparable across birds. Interestingly, though, Jelo exhibited prosocial choices in almost all conditions. If she would not have understood the token values, we would have expected her to choose rather randomly, which was not the case. We have added this information to the manuscript (line 408).*

Furthermore, we found no difference in prosocial choices in the accessible control between birds that experienced the inaccessible condition first and birds that started with the accessible control (Wald Chisq Test: $\chi^2 = 2.67$, $df = 1$, $p = 0.102$). We have added these statistics to the manuscript (line 505ff).

L. 386: What did the birds understand about the blockage? Was this an alternating condition or not? If not alternating, maybe only the bird that didn't get the treat understood the condition and would respond appropriately? Maybe bird A keeps being prosocial because it wants bird B to see that it is trying to be generous no matter what the experimenter does?

Response: *In the social facilitation control, a blockage was placed on the panel in-between partner bird and experimenter. Consequently, the partner bird could neither see the experimenter nor the food rewards. Also, the partner bird did not make any choices during this control but only acted as a passive partner (no alternating choices), like in the unilateral or yoked control condition, however with the important addition that the partner did not receive any rewards.*

Your interpretation is definitely valid, it might be possible that the actor continued to make prosocial choices because he/she did not understand that the partner was not getting a reward. Nonetheless, we observed that the partner birds got frustrated and lost motivation rather quickly during this condition, as they could not gain anything and subsequently retracted to the back of the table instead of staying in the front close to the exchange area. Consequently, we think that the actor birds would have noticed this difference in behaviour (i.e. lack of interest) and could have changed their choice behaviour based on this aspect rather than on an understanding of the blockage.

L. 407ff: The birds that exhibited preferences—were they first actors or first recipients?

Response: *Half of the birds that exhibited token preferences following the test, started as actor (N=3), while the other half started as recipient (N=3).*

L. 411: I think the authors mean S2...

Response: Yes, we wanted to refer to Table S2 instead of S1. We have corrected this mistake in the manuscript (l.457).

L. 437: Again...did the first actor understand the issues? Only when the first recipient became the actor would the task make sense...and if the first actor wasn't very prosocial, the recipient might not be so after it became the actor when there were 30 trials of one type involved...

Response: Certainly, there is the possibility that the attitude (prosocial or selfish) of the first actor influenced the behaviour of the partner when roles were reversed; however, there is no difference in prosocial choices between birds that started as actor or partner (see response above with statistics).

L. 446: Again—possibly the birds were trying to reciprocate with the experimenter in some way, so that it was still some kind of social situation...

Response: The yoked control should not eliminate the social aspect but rather test whether the birds would understand that their choices did not affect the partner's subsequent choices. As stated above, we agree with your interpretation that this condition potentially was rather confusing for the birds. Nonetheless, it is possible that the birds were trying to reciprocate choices with the experimenter and consequently did not reduce prosocial choices in the yoked control. We have added this point to the discussion.

Line 580ff: "Alternatively, it is possible that the parrots were trying to reciprocate choices with the experimenter, as she was the one seemingly selecting the tokens (although she was only copying the choices from the partner in the previous ALT condition)."

L. 457: I'm confused...a partner was present in both UNI and SFC, so why would an equal response suggest that the presence of a partner did not affect their choices? Would have to contrast data between the presence and absence of a partner to make that point...

Response: You are right, the indifference between UNI and SFC actually indicates that prosocial choices might have been caused by social facilitation instead of an understanding of whether help can be provided or not. We changed the sentence accordingly in the manuscript.

Line 500ff: "The birds' prosocial choices did not differ between the UNI condition and the SFC ($49.5 \pm 39.2\%$; GLMM: $\beta = 0.01$, $SE = 0.15$, $z = 0.08$, $p = 0.940$); consequently, we cannot rule out that the presence of the partner facilitated prosocial choices irrespective of whether they actually helped the partner or not."

L. 507: Yes, they were having trouble understanding the task contingencies, but not issues of whether or not a partner was present, absent, or able to access the reward...putting oneself in the situation of the bird, a human might have trouble figuring out the various task contingencies.

Response: We changed the conclusion slightly and now point out that they were having problems with the spatial arrangement rather than the presence/absence of the partner bird.

Line 551ff: "However, the birds did not seem to distinguish between control conditions, in which they could or could not access rewards delivered to both compartments; hence indicating that they were having problems in understanding the task contingencies (i.e. spatial arrangement)."

L. 511ff: But by the time the unequal reward task was administered, the birds had a pretty good idea that turns would be taken, whether immediately or eventually...so they could reason that if they ever wanted the walnut, they'd better give it to their partner...and the effect seems driven mostly by

one dyad...

Response: *As it always is a problem with a limited sample size, single dyads can strongly affect the results; however, when looking at the UNI conditions on the dyadic level (see Table S3), two dyads clearly increased their prosocial choices if unequal rewards were used.*

While we cannot control for an order and thus learning effect, due to the fixed presentation of conditions, we think that your explanation is rather unlikely and that less cognitively demanding behaviours are involved. Considering that roles will eventually be reversed in the future would require quite sophisticated planning into the future. The UNI condition was the first one, in which they were given the possibility to earn a walnut. In all previous conditions they could only receive seeds, consequently, they could have learned that selecting the prosocial token yields more payoff (2 seeds are present vs. 1 seed present) irrespective of whether they actually get these rewards, while the payoff in the unequal UNI condition is even bigger (1 seed and 1 walnut). A more parsimonious explanation is that the simple presence of a walnut might have facilitated the selection of the prosocial tokens in the unequal condition rather than planning ahead through establishing a “generous” attitude.

L. 530: Theoretically the option of tit-for-tat was eliminated, but one would have to show that the synergy was different to make such a claim. It was still a social system in which the birds could have been trying to influence the experimenter...

Response: *This certainly is possible and we have added this explanation to the manuscript (see response above to issue with the yoked control).*

L. 546: Remove comma...sentence unclear....

Response: *We have clarified the sentence: Line 600ff: “The increase in prosocial choices across sessions in the unequal alternating condition could additionally be explained by reinforcement learning (i.e. pick same token as partner) or attitudinal reciprocity.”.*

L. 605: Maybe the problem is not about the birds’ cognitive abilities—the experimenters did not examine social dominance issues or the extent of the ‘bonding’ within the pairs. Dyad 2 here seemed particularly uncooperative; I wonder if they were the least strongly bonded pair or the one with the most difference in dominance...I simply think that the tasks were confusing, and that in some instances other issues such as dominance and the order in which a bird in a pair began a task may have affected the outcomes.

Response: *We agree, relationship quality might have affected the results. Unfortunately, we did not qualitatively assess the social structure within the group prior to the experiment. Nonetheless, dyads were selected based on observations during feeding time and we made sure to pair birds with a good relationship. Interestingly, following this experiment we started to conduct regular behavioural observations within the group and the dyad you are referring to (Lizzy-Nina) actually has a rather good relationship – even better than the two more prosocial dyads (dyad 1 and dyad 3).*

We also do not know how many times the birds simply refused to work...Such behaviour is common in parrots and is not a reflection on the study, but knowing the numbers of failed trials may provide a handle on how confusing the tasks may have been.

Response: *We had to terminate 12 sessions due to birds not being motivated anymore (either to take tokens or to exchange them). Ten of these aborted sessions occurred in one dyad, while the other three dyads participated more reliably. See our response to the comment above. On average no more than 4 failed trials occurred per session.*

I'm very conflicted about this paper—the experimenters tried to design appropriate controls but mainly seemed to have confused their subjects. We know nothing about the number of failed trials and nothing about the actual relationships between the birds in the dyads. The authors are honest in their overall evaluation that additional work is necessary to determine prosociality in parrots (at least Grey parrots), but leave us without much more information than we had from the earlier studies (i.e., that parrots like to play tit-for-tat to some extent, although the previous studies demonstrated that the parrots didn't simply copy the behavior of their partner—at least when the partner was human).

Response: *We disagree, even though the parrots seemingly had problems in understanding the spatial arrangement, several new results emerged from this study: 1) We present results that allow a first direct comparison of parrots' performance in a prosocial choice paradigm with that of previously tested species and that provides important basis for future comparative research, as we closely adhered to the procedure of other studies, mainly in primates. 2) We point out some difficulties this standardized task renders for birds in comparison to primates that can be addressed in the future. 3) The previous results on African grey parrots' prosocial behaviour stem from a single dyad, whereas we provide data on a bigger sample size and on birds that lived in a social group which have only limited and standardized human contact. The claim that we implemented controls that have merely „confused“ the subjects is ungrounded. The only confounding factor which we acknowledge is that the parrots seemed to have difficulties with the spatial arrangement.*

Appendix B

Reply to the reviewer comments:

Associate Editor Comments to Author:

Ordinarily, the journal does not grant authors multiple opportunities to revise their paper(s); however, where 'good faith' attempts at improvement seem to be made by the authors in response to extensive feedback, a degree of flexibility is sometimes permitted. As the Editors consider you to have tried to respond sincerely (if not satisfactorily) to the reviewers, a further opportunity to revise is being made available - you should work hard to tackle constructively the remaining concerns. If you do not do so, and the more critical of the reviewers remains unsatisfied by your efforts, we will not be able to consider further revisions. Good luck.

Subject Editor Comments to Author:

I support the AE's comments and wish you the best in your revisions.

Reviewer comments to Author:

Reviewer: 2

Krasheninnikova and colleagues have revised their extremely interesting paper on prosocial behaviour in Grey parrots. They examine multiple possible situations and test the birds extensively. I nevertheless still have several comments – some involving minor clarification, others a bit more substantive—which I present in order of appearance.

1. L. 29: I'd be more comfortable with something like “..their lack of understanding of the contingencies of the particular tasks used in this study, the underlying...” [that is the most that the authors can claim]

Response: *Thank you for reviewer's suggestion, we have changed this sentence accordingly in Line 29-30.*

2. L. 39: This paper is on pro-sociality, which the authors just described as being different from altruism. So where does this sentence with respect to altruism fit? Do they mean pro-sociality? Or at least reciprocal altruism?

Response: *We thank the reviewer for pointing this out and changed it to “pro-sociality” (l.42)*

3. L. 58-59: The edit is odd...leave out comma after “familiar”...and maybe “and in situations without a possibility...”

Response: *The sentence has been revised according to the comment (ll. 61-62)*

4. L. 118: Remove comma after “While”...Based on the material in the paper, maybe the description should read “did not show clear reciprocity...” Again, read my comments based on the material in the paper cited: When the dominant bird led, he STARTED OUT prosocial but switched to selfish when his partner consistently acted in a selfish manner”...what is written in this manuscript is not correct.

Response: *We have revised the sentence according to the comment (l.121).*

5. This description of the material is not correct. Citing from the paper itself: "However, whether they shared with their partner at no cost to themselves seemed, at least for our subjects, to depend upon their dominance status and upon who led and who followed. Griffin has always been dominant to Arthur, and Figs. 3 and S1 show that when Griffin led, he became selfish, but when placed in the position of follower, he acted in a more sharing manner." That is, when Griffin led he started out sharing but when Arthur became more selfish, Griffin became more selfish. When Griffin followed, he retained his more sharing manner. Arthur, the subordinate, "...started with a slight giving tendency as both leader and follower, then became statistically significantly more selfish both as leader and follower."

L. 120ff: THEREFORE, these lines should read: "If the dominant bird started the session choosing first, he initially chose the prosocial token, but became more selfish after he experienced selfish behavior from his partner. When the dominant bird was the follower, he retained his prosociality. The subordinate bird, however, became more selfish over time in both leader and follower roles. Therefore, both birds developed..."

Response: *We have revised the sentence according to the comment (ll. 122ff).*

7. L. 125: This sentence is not correct—and the sentence has not been altered....I repeat: the authors tested for side biases and the current authors cannot disregard these tests. I will change my decision to "reject" rather than revise if this revision is not made in a subsequent version.

Response: *We apologise for any incorrect wordings in the previous revision. We do not disregard the side bias tests implemented by the original study by Péron et al. (2013) and we now acknowledge it clearly in the text. However, as stated in our previous reply to the reviewer comments, we do have some serious concerns regarding the **efficiency** of that side bias procedure that Péron et al. (2013) carried out. As far as they describe their procedure in their methods, it cannot rule out that the birds develop a side bias subsequently in the test. For example, the authors stated that they "examined the relevance of cup placement on the tray by changing their order randomly...". This statement is, however, not aligned with the materials provided between the main text and the supplementary information because the spatial arrangement of the cups was not consistently and randomly changed. Actually, the cup position was changed only twice across the 24 or 26 sessions that Arthur and Griffin participated in respectively and moreover, those two switches were made in the first half of those sessions (i.e. session 9 or before) of the experiment (it is also not entirely clear if the session number where switches occurred were the same for both birds, as the two arrows indicating the respective cup position switch seem to be misplaced in some graphs (see supplementary information in Péron et al.)). Consequently, since session 9 and for the following 15 (for Arthur) and 17 (for Griffin) sessions the cup order remained the same, which means that a formation of location bias could still have potentially occurred after that last position switch. Moreover, the authors provided statistics that assessed whether the parrots' choices were different from chance and not whether the birds actually preferred one cup over another. Thus, it just cannot be excluded that the birds' performance can be explained by a side bias. Nevertheless, we agree with the reviewer that our previous claim that the birds appeared strongly side biased was phrased too strongly and we have re-phrased this sentence (ll. 131-139).*

8. L. 126: This sentence is also not correct. Control was inherent in the task: Note that the birds did not copy the human, as the birds were not giving treats to the generous human but sharing treats, and

they quickly learned to avoid the null cup (no treats to anyone), demonstrating that they clearly understood the task. Note, too, the follow-up study that further tested the task's assumptions. The authors of the current paper cannot blithely disregard data and criticize another paper without full justification. Avoiding the null cup if it gives no reward can be described any way you choose, but it still involves understanding the contingencies of the task. And explaining the choice of the option that provides two rewards rather than one as reinforcement learning is meaningless unless the chooser GETS more rewards. If the chooser gets to share, that means the chooser understands the contingencies of the choice. As for the control without a partner—the subject is simply likely going to be confused about what is going on—see more on that later.

Response: *We thank for the reviewer's comments, and we have re-studied the study by Péron et al. 2013 and the follow-up study by Péron et al 2014 carefully.*

However, our stance remains the same as outlined in the previous round of revision and we are sorry to disagree with the reviewer on this point. In our view, this behaviour could easily be explained by simple associative reinforcement learning instead of an 'understanding' of the task. The parrots quickly learn to avoid something that is not rewarding for themselves (the null cup) and they preferred to choose an option that provides more rewards (i.e., two pieces under sharing cup). The observation that they avoid the null cup is not sufficient to show that they understand all the task's contingencies but only that the null cup is not reinforced. Also, the fact that the birds were not only avoiding the null cup but also avoiding the giving cup (which means 0/1 distribution, where the actor did not get any reward either) does support the assumption that the birds were just avoiding those options that were not rewarding for themselves. We also wish to highlight that we are not the only ones critical of the study of Péron et al. The methodological limitations have been pointed out before by Marshall-Pescini et al. 2016 and also by Lambert et al. 2019. Still, we have toned down the sentence and now say that "no sufficient control was implemented to assess the parrots' comprehension of the task (see also [24])" (ll.138).

The argument about color bias is justified, however.

9. L. 134: I think the authors mean "employed" rather than "applied"

Response: *Has been changed according to the comment*

10. L. 201: "without shells"

Response: *Has been changed according to the comment*

11. L. 199: This sentence is not clearly written. Try dividing the sentence into two. "...in all equal reward conditions. When the actor chose prosocially in the unequal condition, in contrast, the partner received a piece of walnut (preference for walnut was assessed prior to the experiment) while the actor received merely the same sunflower seed as in the equal condition."

Response: *We thank the reviewer for this suggestion and have changed the sentence accordingly (ll.215ff).*

12. L. 217: I think it is important to note here that these birds did NOT express inequity aversion previously. Such finding suggest that the authors already know that the birds are not that likely to be affected by their partner getting a better reward, and that this is a further test of that finding.

Response: We added a statement that our parrots did not express inequity aversion previously to the discussion (ll. 606-617). However, a lack of inequity aversion does not indicate that the parrots were not caring about which reward their partner received, it rather shows that they did not refuse to cooperate in an event of unequal reward distribution. The fact that the birds tolerated minor inequity, may even increase the likelihood of finding prosocial attitudes as it shows that the birds are very tolerant of seeing their partner receive something better than themselves (see further elaboration on this topic in response to comment 16).

Furthermore, although both studies were based on the exchange paradigm, the methodology differed considerably. E.g., in sharp contrast to the present study, the distribution of rewards could not be influenced by the birds in the inequity study (i.e. the experimenter handed out the rewards) other than by stopping to participate, whilst the actor in our present study could determine the reward distribution (whether it is selfish or prosocial). An additional noteworthy difference is that the current study investigates whether the birds would respond to their neighbor being **disadvantaged** and actively choose an option that would reward both birds equally. In the previous study, in contrast, the situation was reversed, given that both birds were always rewarded by the experimenter, but sometimes the neighbor was **advantaged** compared to the subject and it was measured whether the subject responded negatively to the better treatment of their partner. Thus, the setups were so different from each other, that one cannot use them to make informed predictions about the birds' reaction to an unequal distribution of rewards beforehand. In order to refer to the previous findings but at the same time minimize any confusion, we have added a brief description of the previous inequity study (see lines 236-241) and discuss the finding in the context of this study (ll. 606-617).

13. L. 216: In the response to reviewers, the authors mention that they did not find any position preference in token choice...this information should be provided somewhere in the results section.

Response: We now provide additional information in the method part clarifying this point: "To avoid the formation of location biases during token selection, the tokens (six in total - three of each colour) were jumbled in a small cup and then randomly placed onto a shallow transparent plastic tray (making sure that the tokens did not overlap but still randomly distributed across the tray, i.e. without predefined positions for 6 tokens) before presenting them to the birds." (l.242ff).

14. L. 226: Is this a new paragraph?

Response: Yes, thank you for noticing this. The format has been changed accordingly (l.252).

15. L. 244: Sentence a bit confusing. Divide sentences. "The rewards inside the cups were not at that time visible to the birds. Once a choice was made, the experimenter lifted the lid(s) off the small food container(s), tilted the cup(s) forward so that both birds could see the reward(s) inside, and then distributed the reward(s)."

Response: Has been changed according to the comment (ll.261ff).

16. L. 259ff: If birds don't care about inequality, why would they care if one got fed and one didn't? Isn't that finding a problem for this entire paper? The receiver could eventually associate the ring color with whether it gets food, but not necessarily with the actions of its partner if it doesn't care that its partner gets fed and it doesn't, especially if it was going to get at least half the standard reward just for hanging out near the reward site when the experimenter wanted it to move back. If a whole seed is a standard reward, half that reward is hardly "tiny" [and, yes, I do understand that sunflower seeds cannot be divided into much smaller pieces and still easily be handled, but maybe the authors should have used some other, less favored food?].

Response: Thank you for making this point, we attempt to make this clear in the text now. Yet, we actually do not agree that our previous finding should represent a problem for our entire paper (see also our response 12), but - rather on the contrary-, that the lack of previously found inequity aversion might even bias them to behave cooperatively (-this is something we also argue in this previous study-) therefore making them particularly interesting for an investigation of pro-sociality. As stated above, it rather shows that the birds would tolerate moderate inequity and still cooperate with each other. In fact, Brosnan et al. (2010) found that low levels of inequity can be tolerated while still making prosocial choices. Consequently, if the birds did not show a reaction to unequal treatment and thus seem to be very tolerant to an inequality in the reward distribution, they should be even less restrained to provide food to a conspecific in unequal situations compared to species, which are sensitivity to inequity. We added this point to the discussion (ll.606-617).

Furthermore, the two paradigms have inherent controls that rule out an inability to discriminate between food rewards delivered to the adjacent enclosure and to themselves. In the inequity paradigm, a food control condition was implemented, in which a high-quality reward was delivered to the empty partner enclosure while the actors received only a low-quality reward. Almost all parrots showed a behavioural reaction to the discrepancy. Likewise, in the current study, all parrots needed to pass the training, which ensured that they were familiar with the fact that either a reward was delivered to the partner or only to themselves.

Regarding the additional smaller rewards that have been used for moving the birds back to the starting position, this only happened a few times with some of the birds, thus it is unlikely that this affected the birds' motivation/performance. We have modified the methods description to make this point clearer (ll.273ff).

17. I. 270: 12 sessions needed to be repeated? I'm confused...there are only 9 sessions listed in Table 1... If this statement is not a typo, then there are a LOT of repetitions. Could the authors please let the reader know how many sessions were repeated for each condition? That would help the reader know which conditions were more problematic. The authors gave some of the information in their response but not in the paper...it seems as though one dyad really had issues, and it would be helpful to know which one. Maybe that set of data needs to be dealt with separately?

Response: The statement is correct; 12 sessions have been repeated in total across all dyads. Please note the Table 1 represents how many valid sessions per bird per condition were necessary. That means 18 sessions per dyad, 72 sessions in total. Out of these 72 sessions, 12 sessions were invalid due to different reasons, i.e. lack of motivation or disturbances, and needed to be repeated (l.276). Eight sessions were repeated in the alternating condition (for the same dyad), and one session each of Inaccessible-Equal, Accessible-Equal, Social Facilitation, and Accessible-Unequal across two different dyads. The lack of motivation in the dyad Nikki-Jack was mostly due to Nikki who was gaining weight and not being food motivated anymore preferring to play with the token rather than handing it back for food. After 4 days of pause in the testing in the alternating condition for this dyad and adjusted amount of daily food, Nikki reduced her weight and was motivated again. Two more adjustments of Nikki's daily food amount in the course of the study were needed to keep her motivated. Therefore, we do not think that the data from this dyad needed to be dealt with separately. Two sessions were invalid because of disturbances caused by maintenance work outside the laboratory. We now provide this information in the Supplementary Information (see Table S3).

18. L. 273: What happened after the birds chose a token in the color preference test? Did it just get taken away? Did they get something or nothing for making the choice? If they had any experience in trading tokens for treats, and they chose a token and got nothing, they would—if they had any

sense—alternate choice of tokens to see if choosing a different token would give a reward, even if they liked their first choice. It seems that most birds started with a color preference and those colors were eliminated, but it also seems possible that the birds just LEARNED at this point that they should choose randomly because nothing would happen almost whatever they did BUT if they chose their favorite it would disappear. I don't disagree that one has to check for color preferences, but it's a rather tricky procedure as they learn to ignore any contingencies connected with the color.

Response: *Thank you enquiring about those details, thus pointing out that we did not make this sufficiently clear. In the colour preference test, any choice was rewarded in the same way. Two different tokens were presented in one session in a shallow box. After the bird chose one token, it was rewarded with a sunflower seed and the trial ended. The next trial started after a 10 sec interval. Each subject was offered 12 choices, and if any token was chosen 10 times or more by one of the two birds of one pair, the token would be replaced by another colour in the next session. Hence, we did not discourage the birds from choosing particular colours, but just removed colours that we found to be preferred thus biasing their decisions. In the revised version, we have added more details and made this procedure clear (ll. 284ff).*

19. L. 286ff: So now a bird was given "x" to choose and got a reward half the time and was given "y" to choose the other half and still got a reward; sometimes their partner got a reward and sometimes the partner did not. Given that they didn't care about reward inequality, why would they pay much attention to what was happening, particularly if they were actor but even if they were the powerless recipient? If the recipient couldn't do anything about what happens, and can't tell the actor what to do, why pay too much attention?

Response: *As elaborated above in our replies to comment 12 and 16, we do not think that our previous finding (that the birds did not show inequity aversion), does in any way suggest that the birds do not pay attention to or do not care about what their neighbour receives. Our controls show that they do pay attention to what food is delivered to the other side, but they do not display a negative response if the partner receives a better treatment. As pointed out before, it is not conceivable that not reacting negatively to a conspecific receiving an advantage, should constrain prosocial behaviour (see also Brosnan et al. 2010. "Competing demands of prosociality and equity in monkeys", explaining effects of inequity on prosociality), where one takes action so that another individual is not disadvantaged (in the sense that it otherwise receives nothing). We also would like to point out again in this context -because that could be the source of a recurring misunderstanding between reviewer 2 and us - that in our previous inequity aversion study we have not tested the subject's response to their partner receiving an inferior reward or no reward at all, which would also be unequal but disadvantaging the partner rather than the subject. In the current study however, exactly this is happening – the partner either receives a worse treatment (=unequal treatment), namely nothing, or the same food (=equal treatment), whereas the subject always receives food. The biggest difference between our previous study on inequity aversion, where the subject has no control over the reward distribution, the current study is that in the pro-sociality study, it is the subject who decides as to whether to provision its partner with food or not, hence about inequity occurring or not.*

In reply to the reviewer's last 2 sentences, we have reason to believe that the subjects did pay attention to what happened in the other compartment, and we have also evidence that even recipients pays attention to the reward contingencies in both compartments as revealed by the increase in prosocial choices in our ALT (alternating) condition. In the ALT condition, the recipients were not "powerless" because they had the opportunity to reciprocate/retaliate in an alternating manner.

20. L. 298: And I'm still confused...The authors mention equal and unequal rewards earlier, but Table 1

does not, so I'm confused. Does each bird have one session as actor with equal rewards then one with unequal reward before the roles are reversed (i.e., that's what is meant by 2 sessions in Table 1)? Or do reversals occur after equal reward and then again before unequal reward? And do all these trials occur before the alternating sessions? I need to understand what is happening at this point in the paper. Or do the birds cycle through UNI, ALT, YOK in the equal reward before doing unequal rewards? [It seems that way, given that different tokens were used, but I really had to work to figure this out and I'm still not sure.] If I were trying to replicate this experiment, I'd never be able to figure out exactly what was done. The material now taken out helped me before...

Response: *We thank the reviewer for pointing out that the procedures have been hard to understand. We have made some improvements in the revised text (L.335ff) and added a Table in the supplementary Information (Table S5) with an example of how the conditions were presented to the birds. In answer to your question, yes, the birds cycled through UNI, ALT, YOK with equal rewards before being tested again with unequal rewards. We hope this does become clear to the reader now.*

21. L. 305: Two sessions EACH? Including equal or unequal rewards?

Response: *Two sessions each for the equal and two sessions each for the unequal rewards were conducted. We clarified it now in the main text (ll.333ff). We have also revised Table 1 to make this information clearer.*

22. L. 343: Do the authors mean 30 trials per bird acting as actor and 30 as recipient? And where do the equal/unequal fit in here...one session with equal rewards, one session with unequal rewards? Not clear...

Response: *As outlined in ll. 333ff and as now shown in the Table S5 we conducted 60 trials per bird acting as actor and 60 trials acting as recipient per condition for equal reward distribution and another 60 trials per bird acting as actor and 60 trials acting as recipient per condition for unequal reward distribution.*

23. L. 355ff: I absolutely do not understand the authors' response to my query. Why would switching to a selfish strategy be the expected response? The experimenter is NOT always being selfish, so why would the actor be selfish? The authors are making AN ASSUMPTION that the birds are treating the experimenter as an automaton, and ignoring the possibility that they were playing with the experimenter. In previous studies, didn't the experimenter played partner-like roles? So why ignore that now? Given that there were lots of prosocial choices in the alternating condition, the experimenter was being prosocial a lot, so why wouldn't the actor reciprocate to that? I definitely appreciate the new material added around L. 580.

Response: *We are sorry, but we do not understand this comment. We do not say that switching to a selfish strategy is the expected response, but that there should be a difference between the yoked control and the alternating condition. In any case, we don't understand why the reviewer suggests that the experimenter behave prosocial in the alternating condition – she did not: in this condition the birds interacted with their partner who made the choices not with the experimenter. Also, in our previous studies the experimenter did not play partner-like roles, so we are not sure what we may be ignoring.*

24. L. 384: Accessible vs inaccessible: So, I'm a bird and you train me to go back and forth between compartments. I learn that I can get food in both places, but not with respect to anything to do with token choice; I just learn that it's possible. And there's no bird next door. If I'm Jelo or Bella or Sensi or Nikki (half the birds), I don't notice whether or not the passage is open but I choose to put food on the

other side because I remember that it's possible to get food. If I'm Kizzi or Nina or Jack, whether or not the passage is open I'm not prosocial maybe because there's no bird there. If I'm Kimmi, I'm just confused. Actually, if the birds differentiated between the conditions, they should NOT be random but choose selfish in inaccessible because they'd otherwise be wasting food, and they should choose prosocial in the accessible condition to get more food. But none of the birds (except a tendency in Jack, in the opposite to what would be expected) seemed to understand the different conditions at all. They seemed to hit on some strategy simply with respect to the possibility of more food—hope for the best or accept the worst—and stick to it. I don't see the task as showing much of anything about understanding cooperation. I think the authors' comments in the Discussion about what they did and didn't learn and spatial confusion make more sense than the assumption that the task would be useful, particularly given that the birds' behaviours altered over time.

Re blockage...about half the birds seemed to understand the issue and the other half did not.

Overall: Individual differences seem to be extremely important in this study, from looking at the supplemental data. Maybe some Greys are good buddies and others simply are not, or maybe some birds are dim and others not...the authors really need to look at and discuss the individuals, not just the group behaviour.

Response: *We agree with the reviewer that the individual variation is extremely important in cognitive studies. However, the goal of our study is to assess whether adding the control conditions (alongside increasing the number of dyads tested) would allow us to detect the parrots, **at group level**, show pro-sociality. Hence, we focus more on the forest than the trees or, not solely on individual variation. Nevertheless, we do mention individual differences throughout the manuscript. We have also added information about some individuals that appeared to understand the test more than the others, or not least, to be more prosocial than the others (supported by the results of the UNI condition) but we are careful not to make purely speculative interpretations. Additional information about individual differences are also provided in supplementary information (Table S3), which readers can access if they are interested in looking into this further. In the SI we have also added some tentative suggestions from those individual data, but, in our opinion, such interpretation of individual data is of too speculative nature for the main text. In other words, we can also merely speculate on how this individual variation is explained, i.e. whether behaved more prosocial some because they were better 'buddies' or did not understand the controls because they were more 'dim' than others (no visible from the observational data or from prior cognitive performance).*

25. L. 451ff: I'm confused...if birds DID develop preferences, and there were so many individual differences in the results, shouldn't the experimenters have checked specifically to see if the birds' actions were affected by these preferences? Just looking at group behaviour is not sufficient, as the individuals varied so strongly.

Response: *For reasons outlined in the reply to the comment 24, the focus of our study was on the group level performance.*

26. L. 474: The authors must mean dashed red line, not dotted.....I'm not sure what the dotted line means.

Response: *We thank the reviewer for noticing this. We corrected it to: "The dashed red line....." (L.462).*

27. L. 560: Correct English is "fewer prosocial", not "less prosocial"...

Response: *Has been changed according to the comment.*

28. We still do not know the relationships between the individuals in the pairs, although the authors do admit to the artificiality of the situation. Note: A 'good' relationship doesn't necessarily mean equal; if the birds have worked out a dominant-subordinate relationship and are happy with it, that makes it "good"...but if there is conflict, that is a different situation, and could easily affect the results.

We also know that parrots engage in courtship feeding, where one bird gives up food to the other with no reciprocity, and in Greys both parents feed the young directly from what they obtain when they forage (unlike mammals, who suckle, which is quite different in terms of sharing resources)...so the whole idea of unequal reward as an issue is quite different from what one would see in primates.

Response: *We thank the reviewer for this feedback. We do agree with the reviewer that 'good' does not mean 'equal'. We defined 'good' as birds that showed more affiliative behaviours towards another bird than others, regardless of sex. This aims to increase the probability to detect whether the birds understand pro-sociality. While the parrots in the partially natural set up of the zoo do exhibit courtship behaviours, the parrots under this set up are prevented from breeding. This follows that we do not have parents, young or the need of feeding the young by any birds. Nevertheless, birds that paired up in this study are either sibling (e.g. Jack and Nikki) or same-sex birds that are 'friends' with each other (which they could not breed). We did not collect quantitative data about the birds' dominance relationship for this study.*

29. I'm still very conflicted about this paper—given how critical the experimenters are of the previous studies, their failure to examine the material of the previous studies carefully, and their claims that they performed their experiments in order to correct all the problems they perceived in the previous studies, they really have to be exceptionally careful about what they are saying and claiming. I re-read the paper even more carefully this time, and found other inconsistencies that need to be addressed. I'm not trying to be obnoxious—I don't want someone else to take them apart after the paper is published. I think that looking at the behaviour both of individuals and individual pairs would be very helpful.

Again, the good part of this paper is that the authors are honest in their overall evaluation that additional work is necessary to determine prosociality in parrots (at least Grey parrots). And, yes they did test more birds than in previous studies, but looking at the individual data it seems that some birds are good at being prosocial, others are not, and some learn more about the situation than do others. And I repeat that their controls were confusing...if I had been in the birds' place, I would have been confused. Just because someone else used these controls doesn't mean they are correct. Thus, again, the authors have to be extremely careful about their analyses and claims as well as criticisms.

Response: *We are grateful that the second reviewer has provided many helpful comments and suggestions for improving the manuscript and we have taken them very serious and have implemented them wherever we could. Yet, we feel that this last comment is neither constructive nor fair – reducing our study as a failed attack of a previously published study with a different method, and completely neglecting the value of our overall results at group level. While we do value data on individual variation and agree that they can reveal a lot and inspire future studies with bigger sample sizes, we admittedly find it very odd that reviewer 2 seems to value individual data higher than group level performance.*

Given that reviewer 2 keeps stressing that we are being unduly critical of previous work on grey parrots, we have toned down our critical remarks about that study in our objectives, of course our goal was not to devalue the previously published work but to build on it.

Concerning the suggestive comment about preventing embarrassment later, we are actually fairly confident that given the published papers on primates (de Waal et al. 2008, Suchak et al. 2012) which we wanted to compare our data against, our manuscript would not be torn apart if published in the current form and we are not aware of any further inconsistencies that reviewer 2 mentions but fails to specify.

Concerning a stronger focus on the individual data, we counter that any reader who is interested in the individual data which we provide in the online supplementary information, is free to look at them and draw their own conclusions (or conduct meta-analyses together with other larger datasets), yet we do not feel that it is productive to dwell on individual data in order to draw speculative conclusions.

Concerning the comment about the supposed confusion of the bird; it is the overarching dilemma of animal cognition studies, that we do not know how the test animals under investigation perceive a test situation. Most studies aim at designing their studies as ethologically valid and salient for the species under investigation as possible, but often it turns out retrospectively that unintended variables affected the animals in the test situation. This is a common pitfall of comparative cognitive studies in particular where one strives for maximum comparability by keeping the experimental protocols as similar as possible. In our case, we aimed at comparing the performance of parrots to that of monkeys, and therefore implemented as few adaptations to the avian model as possible. We acknowledge and discuss that our birds seemed not to have fully grasped the contingencies of the controls. Yet, it is not the case, as reviewer 2 seems to suggest, that one could have predicted that the control would have been too confusing for the birds. On the contrary, we are still convinced that the rationale behind the non-social control was a reasonable one and might work in other parrot species or with more familiarity with the physical setup.

Reviewer: 1

Comments to the Author(s)

I sincerely thank the authors for their thoughtful responses to my earlier concerns. At this point I have reviewed the revised manuscripts and have no further questions or concerns to raise.

Response: *We thank the reviewer for this feedback.*

Appendix C

Associate Editor Comments to Author:

The reviewer has presented a number of critiques of this paper through its iterations, and the Editors would like to thank them for the guidance and also encourage the authors to constructively engage with these critiques. Please do ensure that you take the time to make the changes requested, and present these in a tracked changes version of the paper, along with a full point-by-point response. This will help the Editors determine whether the reviewer needs to be consulted further.

Reviewer comments to Author:

Reviewer: 2

Comments to the Author(s)

Krasheninnikova and colleagues have re-revised their extremely interesting paper on prosocial behaviour in Grey parrots. They examine multiple possible situations and test the birds extensively. I nevertheless still have several comments – some involving minor clarification, others a bit more substantive—which I present in order of appearance.

L. 75: Authors might be interested in a new paper: PLoS One: Female rats release a trapped cagemate following shaping of the door opening response: Opening latency when the restrainer was baited with food, was empty, or contained a cagemate

Magnus H. Blystad , Danielle Andersen, Espen B. Johansen

Reply: *We thank the reviewer for this suggestion, however, we believe that this study is not relevant for this paragraph, as we refer to a review that summarizes different paradigms already.*

Also: Should read “methodologies”

Reply: *Changed according to the comment.*

L. 104: Odd grammar: Probably should read something like “Although few studies have focused on...in the wild, a large number of studies...in the laboratory.”

Reply: *Changed according to the comment.*

L. 124: The authors again do not attend to the studies. They are confusing ref. 53 and ref. 54. In a second experiment in *ref. 53*, the human was either generous or selfish and the parrots modulated their responses accordingly. In the follow-up study in *ref. 54*, the human exactly mirrored the parrot’s behavior, and the parrot learned the “tit-for-tat” response quickly. So...if the authors want to ignore the human-parrot study (ref. 54) to some extent because it is not parrot-parrot, that is a separate issue, but they should not confound the two studies....

Reply: *We thank the reviewer for this feedback. In the successive revisions, we have indeed, by mistake, confused the references. We have rectified this and now explain both the original and follow-up study studies better (see next comment).*

L. 129: This description of the material is STILL not correct. Again, I am truly annoyed at the authors’ inability to separate the second experiment in ref. 53 from the study in ref. 54. In ref. 54, citing from the paper itself: “Cup positions were now changed after every human choice, with positions determined by use of the program random.org.”

Reply: *We have clarified this in the current revision.*

The point is that the authors did check for side biases in their subsequent study...and there were NO effects... If the same bird did not have position preferences in ref. 54, it did not have them in ref. 53.

Reply: *We disagree, the parrot individual in [54] was tested three and four years, respectively, after it had participated in [53]! The fact that this bird did not have position preferences in ref. 54, does not prove that it did not have them in ref. 53, also considering all the experiments it might have been involved in in the meanwhile. Also, unfortunately, the second bird could not be retested in 54, so that no conclusions about his potential side bias can be made.*

According to the authors in ref. 54, that subsequent study was also designed to examine the parrot's understanding of the task...would the parrot learn that the recipient was playing tit-for-tat and respond in kind? So, again the statement with respect to understanding is not valid, and the citation of ref. 24 also is with respect to ref. 53 (ref. 24 does not even cite ref. 54, where the bird was the first actor).

Reply: *We have clarified this in the current revision.*

The argument about color bias is justified. The current authors maybe should just stop there, or clarify that their problem with refs. 53 and 54 was that no control existed to determine how the subjects would act in the absence of partners.

Reply: *We have rephrased this sentence (l.133-135).*

L. 139: The third objective is the same as in the second experiment of ref. 53 and was central to ref. 54.

Reply: *Yes, however, the reciprocity in the two aforementioned studies was in regard to a human experimenter, whereas in the present study the focus was on the reciprocity towards a conspecific. It has already been pointed out by early studies in comparative cognition that results may differ when animals are interacting with human experimenters rather than conspecifics (e.g., Hare, B., Call, J., & Tomasello, M. (2001). Do chimpanzees know what conspecifics know? *Animal behaviour*, 61(1), 139-151). We clarified this point in the current revision (l.141ff).*

L. 146: Grammar—"We varied the reward distribution, using "equal" or "unequal" rewards, to test....distribution (i.e., making....)" [I believe a parenthesis is missing]

Reply: *Changed according to the comment.*

L. 154: "...chose the prosocial..."

Reply: *Changed according to the comment.*

L. 198, 199: "When the actor..."

Reply: *Changed according to the comment.*

L. 226: "...overlap but were still..."

Reply: *Changed according to the comment.*

L. 253: How many is "a few"?

Reply: *This happened in all birds in around 10 % of the trials. We provided this information in the current revision (l.257f).*

L. 268: The authors' response to the reviewer comments was a lot clearer than what they wrote in the manuscript... In the colour preference test, any choice was rewarded in the same way. Two different tokens were presented in one session in a shallow box. After the bird chose one token, it was rewarded with a sunflower seed and the trial ended. The next trial started after a 10 sec interval. Each subject was offered 12 choices, and if any token was chosen 10 times or more by one of the two birds of one pair, the token would be replaced by another colour in the next session. This additional information about reward would be extremely helpful, so that readers understand what the bird actually experienced.

Reply: *We thank the reviewer for this feedback and have provided additional information according to the comment.*

L. ~276ff: In the familiarization condition, a bird was given "x" to choose and got a reward half the time and was given "y" to choose the other half and still got a reward; sometimes their partner got a reward and sometimes the partner did not. I repeat...why would they pay much attention to what was happening, particularly if they were actor but even if they were the powerless recipient? If the recipient couldn't do anything about what happens, and can't tell the actor what to do, why pay too much attention? That's what happened in the UNI condition...The authors' response makes it clear that the birds care in the ALT condition, in which they are NOT powerless:In reply to the reviewer's last 2 sentences...we have also evidence that even recipients pays attention to the reward contingencies in both compartments as revealed by the increase in prosocial choices in our ALT (alternating) condition. In the ALT condition, the recipients were not "powerless" because they had the opportunity to reciprocate/retaliate in an alternating manner.....which is exactly my point...there is no reason to pay attention unless they can take an active role.

The authors have to clarify WHY the UNI condition is useful earlier on (they do so in L. 307, but until then a reader likely thinks it's a totally useless task)... In the Discussion, they should emphasize the point that in the UNI condition, birds' failures to respond as expected could have been because they did not receive feedback from the recipient, who was powerless to respond, and because the recipient might not even have attended to the behaviour of the actor in the UNI condition for that reason.

Reply: *We thank the reviewer for this comment and have clarified the sense of the UNI condition, namely, to test for spontaneous tendency to benefit a partner earlier in the text (Introduction, L.153). Furthermore, we added a sentence to the discussion according to the reviewer's comment (ll. 526ff).*

L. 327ff: The authors seem not to understand my criticism. My point: The authors are making AN ASSUMPTION that the birds are treating the experimenter as an automaton, and ignoring the possibility (at least until the Discussion) that they were playing the game with the experimenter. The issue is not only whether the actor understood that the recipient could not choose, but also UNDERSTOOD that the experimenter was NOT deliberately choosing...How could the authors be sure of the parrot's interpretation? They suggest this possibility in the Discussion, but it needs to be made clearer.

Reply: *We understood the reviewer's point, however, as outlined in the previous revision round, our birds are not used to an experimenter playing partner-like roles, thus it is unlikely that they suddenly start perceiving the experimenter as a partner. Nevertheless, we had acknowledged this alternative explanation already in the previous revision round (ll. 559ff).*

L. ~360ff: Again, the authors seemed not to understand my criticism....I'm basically questioning the usefulness of Accessible vs Inaccessible: So, I'm a bird and you train me to go back and forth between compartments. I learn that I can get food in both places, but NOT with respect to anything to do with TOKEN choice; I just learn that it's possible when, for some odd reason, there's no bird next door. If I'm Jelo or Bella or Sensi or Nikki (half the birds), I don't notice whether or not the passage is open but I choose to put food on the other side because I remember that it's possible to get food. If I'm Kizzi or Nina or Jack, whether or not the passage is open I'm not prosocial maybe because there's no bird there. If I'm Kimmi, I'm just confused. None of the birds (except a tendency in Jack, in the opposite to what would be expected) seemed to understand the different conditions at all. They seemed to hit on some strategy simply with respect to the possibility of more food—hope for the best or accept the worst—and stick to it. I don't see the task as showing much of anything about understanding cooperation. Or...Maybe the birds kept hoping their actions would eventually help their partner, whom they expected to return because the partner had been there for all the other types of trials?

Reply: *We agree with the reviewer that these conditions might be confusing and discuss extensively possible explanations for the parrots' behaviour in the current study. However, it cannot be ignored that previously a number of species obviously understood the test contingencies and responded according to the predictions, thus it was reasonable to test the parrots with the same set-up as well.*

What about a different control scenario that would not be confusing to the bird?...What if the recipient had plenty of food and didn't NEED anything from the actor, would the actor have continued to be prosocial?

Reply: *As outlined in the previous revision round, we acknowledge that utilizing a more intuitive set-up might have had effect on the parrots' performance (l.617ff). While your suggestion for a control condition sounds interesting, we think that this would add further confounding variables, such as 1) individual food tolerance – does the bird allow another bird close to his/her food? And 2) inhibitory control – can they refrain from begging but instead focus on selecting a token despite their partner eating in the neighbouring compartment?*

L. 395: Again, I see this task as simply confusing to the birds, not an actual test of their understanding...

Reply: *Given that our study used a sound method that is directly comparable to previous primate studies, it adds important comparative data to the phylogenetic comparison of prosociality, even if one of the conclusions is that this particular method bears some problems when used in birds. Again, we are convinced that the rationale behind the non-social control was a reasonable one, also considering previous findings in other species, and that might work in other parrot species or with more familiarity with the physical setup. We acknowledge and discuss extensively that our birds seemed not to have fully grasped the contingencies of the controls (ll.594-633).*

L. 511: "...although such tendencies..."

Reply: *Changed according to the comment.*

L. 515ff, 586ff, 642: The issue to me is that the task contingencies are inherently confusing...which is the point of my criticisms above. I understand that the authors wanted to replicate work with nonhuman primates, but I simply don't think that some of the tasks actually tested what they were purportedly designed to test. Parrots may be inherently less competitive than most nonhuman primates; tasks that might make sense to nonhuman primates might not make sense to parrots...Again, I understand that the authors wanted direct comparisons with nonhuman primates, but then they need

to spend more space in the Discussion (not just a brief mention) explaining why their data differed, based on the ecology and ethology of parrots compared to nonhuman primates.

Reply: *Given that not only nonhuman primates but also dogs have been tested with a similar set-up, it is unlikely that it is a "primate-specific" set-up. We agree that considering parrots' ecology when interpreting the results would provide a more comprehensive picture of parrots' behaviour in the experiment. However, as the Rev 2 is probably very well aware of, there is surprisingly little knowledge about the ecology of this avian group so that any discussion based on the ecology of parrots compared to nonhuman primates is highly speculative and we would like to refrain from such speculations.*

And, as some dyads were just more prosocial than others...one of the in-depth issues for the Discussion is the possibility that individual relationships may be more important to parrots than nonhuman primates and that trying to examine the issue on a group level may be more difficult than it might be in nonhuman primates...That is, maybe if these individual relationships are so important, the birds will *respond* in ways that differ from how nonhuman primates respond?

Reply: *We thank the reviewer for this comment and added a text passage dealing with this point in the current revision (L.659ff).*

L. 550: Yes...

Reply: *We are sorry but we do not understand this comment.*

L. 557: Maybe the UNI prosocial walnut choice was some attempt to get the receiver to attend to what was happening? Given that the UNI unequal followed the ALT equal, maybe they thought that somehow if the recipient got a special treat, s/he would respond, breaking the UNI condition?

Reply: *Please see Table S5 for an example of the order of the testing conditions. The UNI unequal did not follow the ALT equal but was carried out after all conditions with an equal reward distribution were finished.*

L. 668: I don't think the issue was the lack of cognitive resources in the parrots as much as the confusing conditions of the task.

Again, the good part of this paper is that the authors are honest in their overall evaluation that additional work is necessary to determine prosociality in parrots (at least Grey parrots). And, yes they did test more birds than in previous studies, but looking at the individual data it seems that some birds are good at being prosocial, others are not, and some learn more about the situation than do others. And I repeat that their controls were confusing...if I had been in the birds' place, I would have been confused.

What I'm trying to get at here is that the authors made an excellent, good-faith attempt at using a protocol that was designed for nonhuman primates, with very limited revision, for Grey parrots. They found that when the birds could *actually* interact with one another, some pairs acted prosocially, other pairs (like the pair in ref. 53), did not. They also found that the controls may not have been appropriate for the parrots...yes, the tasks might have been beyond the parrots' processing abilities, but given how many other tasks these birds solve appropriately, it is more likely that the birds "overthought" the tasks and responded in ways more appropriate to their ecology/ethology.

In sum....A lot of what the authors wrote in response to the reviewer comments would be a terrific part of the Discussion, and I hope that the editors allow the authors another round so that they can insert

that material. My only disagreement with the following is “unintended variables”—the issue is more likely different types and degrees of species-specific social interactions in parrots and nonhuman primates...the authors almost get to it in the current version, but what they write in their rebuttal would be a lovely end to their paper: *it is the overarching dilemma of animal cognition studies, that we do not know how the test animals under investigation perceive a test situation. Most studies aim at designing their studies as ethologically valid and salient for the species under investigation as possible, but often it turns out retrospectively that unintended variables affected the animals in the test situation. This is a common pitfall of comparative cognitive studies in particular where one strives for maximum comparability by keeping the experimental protocols as similar as possible.*

Reply: *We thank the reviewer for this constructive feedback and added the suggested text passage to the discussion (l.679-685).*